# Inferring characteristics of bacterial swimming in biofilm matrix from time-lapse confocal laser scanning microscopy

Guillaume Ravel[1,2], Michel Bergmann[3,4,5], Alain Trubuil[6], Julien Deschamps[7], Romain Briandet[7], Simon Labarthe[1,2,6]*

[1]University of Bordeaux, INRAE, BIOGECO, Cestas, France; [2]Inria, INRAE, Talence, France; [3]Memphis Team, INRIA, Talence, France; [4]University of Bordeaux, IMB, UMR 5251, Talence, France; [5]CNRS, IMB, UMR 5251, Talence, France; [6]Université Paris-Saclay, INRAE, MaIAGE, Jouy-en-Josas, France; [7]Université Paris-Saclay, INRAE, AgroParisTech, Micalis Institute, Jouy-en-Josas, France

*For correspondence:
simon.labarthe@inrae.fr

Competing interest: The authors declare that no competing interests exist.

**Abstract** Biofilms are spatially organized communities of microorganisms embedded in a self-produced organic matrix, conferring to the population emerging properties such as an increased tolerance to the action of antimicrobials. It was shown that some bacilli were able to swim in the exogenous matrix of pathogenic biofilms and to counterbalance these properties. Swimming bacteria can deliver antimicrobial agents in situ, or potentiate the activity of antimicrobial by creating a transient vascularization network in the matrix. Hence, characterizing swimmer trajectories in the biofilm matrix is of particular interest to understand and optimize this new biocontrol strategy in particular, but also more generally to decipher ecological drivers of population spatial structure in natural biofilms ecosystems. In this study, a new methodology is developed to analyze time-lapse confocal laser scanning images to describe and compare the swimming trajectories of bacilli swimmers populations and their adaptations to the biofilm structure. The method is based on the inference of a kinetic model of swimmer populations including mechanistic interactions with the host biofilm. After validation on synthetic data, the methodology is implemented on images of three different species of motile bacillus species swimming in a *Staphylococcus aureus* biofilm. The fitted model allows to stratify the swimmer populations by their swimming behavior and provides insights into the mechanisms deployed by the micro-swimmers to adapt their swimming traits to the biofilm matrix.

## Editor's evaluation

This paper nicely considers how the biofilm matrix impacts the organism's moving within that environment, connecting prior analyses of cell movements on/within abiotic substrates to those within a "living" substrate. Though there are instinctive descriptions for this motility, the strength of this manuscript is the development and implementation of a statistical model that quantifies critical parameters and incorporates interactions with the biofilm matrix itself. While the manuscript measures the differences between morphologically distinct bacteria, a long-term possibility is to achieve predictable and reliable delivery of antimicrobials (delivered by bacteria or an abiotic object) into the biofilm's center, thereby reducing a biofilm's recalcitrant responses to biocontrol chemicals.

**eLife digest** Anyone who has ever cleaned a bathroom probably faced biofilms, the dark, slimy deposits that lurk around taps and pipes. These structures are created by bacteria which abandon their solitary lifestyle to work together as a community, secreting various substances that allow the cells to organise themselves in 3D and to better resist external aggression.

Unwanted biofilms can impair industrial operations or endanger health, for example when they form inside medical equipment or water supplies. Removing these structures usually involves massive application of substances which can cause long-term damage to the environment.

Recently, researchers have observed that a range of small rod-shaped bacteria – or 'bacilli' – can penetrate a harmful biofilm and dig transient tunnels in its 3D structure. These 'swimmers' can enhance the penetration of anti-microbial agents, or could even be modified to deliver these molecules right inside the biofilm. However, little is known about how the various types of bacilli, which have very different shapes and propelling systems, can navigate the complex environment that is a biofilm. This knowledge would be essential for scientists to select which swimmers could be the best to harness for industrial and medical applications.

To investigate this question, Ravel et al. established a way to track how three species of bacilli swim inside a biofilm compared to in a simple fluid. A mathematical model was created which integrated several swimming behaviors such as speed adaptation and direction changes in response to the structure and density of the biofilm. This modelling was then fitted on microscopy images of the different species navigating the two types of environments.

Different motion patterns for the three bacilli emerged, each showing different degrees of adapting to moving inside a biofilm. One species, in particular, was able to run straight in and out of this environment because it could adapt its speed to the biofilm density as well as randomly change direction.

The new method developed by Ravel et al. can be redeployed to systematically study swimmer candidates in different types of biofilms. This would allow scientists to examine how various swimming characteristics impact how bacteria-killing chemicals can penetrate the altered biofilms. In addition, as the mathematical model can predict trajectories, it could be used in computational studies to examine which species of bacilli would be best suited in industrial settings.

## Introduction

Biofilm is the most abundant mode of life of bacteria and archaea on earth (*Flemming and Wuertz, 2019*; *Flemming et al., 2016b*). They are composed of spatially organized communities of microorganisms embedded in a self-produced extracellular polymeric substances (EPS) matrix. EPS are typically forming a gel composed of a heterogenous mixture of water, polysaccharides, proteins, and DNA (*Flemming et al., 2016a*). The biofilm mode of life confers to the inhabitant microbial community strong ecological advantages such as resistance to mechanical or chemical stresses (*Bridier et al., 2011*) so that conventional antimicrobial treatments remain poorly efficient against biofilms (*Bridier et al., 2015*). Different mechanisms were invoked such as molecular diffusion-reaction limitations in the biofilm matrix and the cell type diversification associated with stratified local microenvironments (*Bridier et al., 2017*). Biofilms can induce harmful consequences in several industrial applications, such as water (*Beech and Sunner, 2004*), or agri-food industry (*Doulgeraki et al., 2017*), leading to significant economic and health burden (*Köck et al., 2010*). Indeed, it was estimated that the biofilm mode of life is involved in 80% of human infection and usual chemical control leads to serious environmental issues (*Bridier et al., 2011*). Hence, finding efficient ways to improve biofilm treatment represents important societal sustainable perspectives.

Motile bacteria have been observed in host biofilms formed by exogenous bacterial species (*Houry et al., 2012*; *Li et al., 2014*; *Piard et al., 2016*; *Flemming et al., 2016a*). These bacterial swimmers are able to penetrate the dense population of host bacteria and to find their way in the interlace of EPS. Doing so, they visit the 3D structure of the biofilm, leaving behind them a trace in the biofilm structure, that is a zone of extracellular matrix free of host bacteria (*Figure 1a* and *Appendix 1— figure 3*). Hence, bacterial swimmers are digging a network of capillars in the biofilm, enhancing the diffusivity of large molecules (*Houry et al., 2012*), allowing the transport of biocide at the heart of the biofilm, reducing islands of living cells. The potentiality of bigger swimmers has also been studied

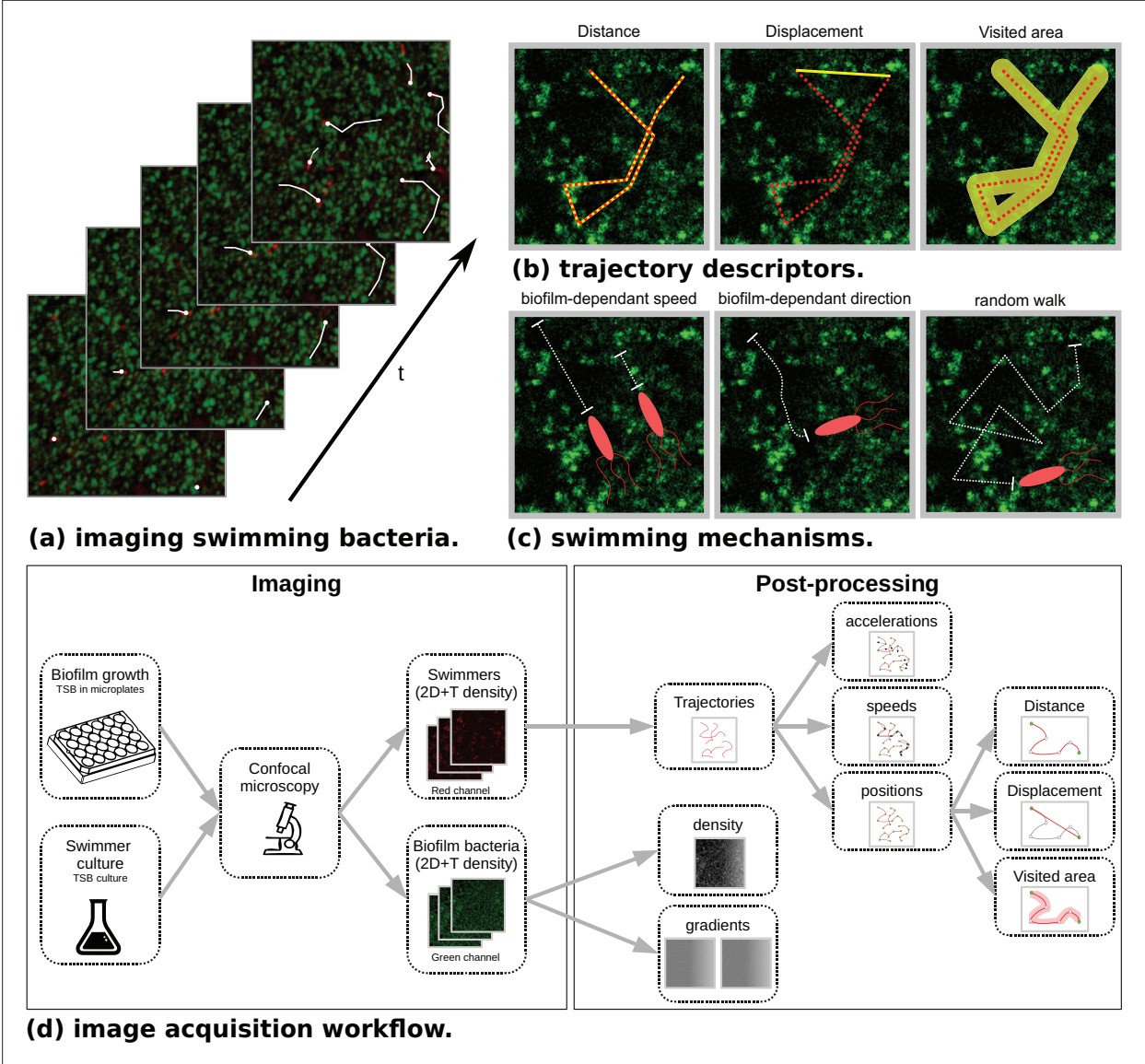

**Figure 1.** Microscopy data and model outlines. (**a**) Temporal stacks of 2D images are acquired, with different fluorescence colors for host bacteria (*Staphylococcus aureus*, green) and swimmers (*Bacillus pumilus*, *Bacillus sphaericus* or *Bacillus cereus*, red). Bacterial swimmers navigate in a host biofilm and are tracked in the different snapshots. Swimmer trajectories are represented with white lines. High density and low density zones of host cells are visible in the biofilm (green scale). (**b**) Additionally to speed and acceleration distributions, three trajectory descriptors are considered. *Distance* is the total length of the trajectory path. *Displacement* is the distance between the initial and final points of the trajectory. *Visited area* is the total area of the pores left by the swimmer during its path. Hence, when a swimmer retraces its steps, the displacement is incremented but not the visited area. (**c**) Three different mechanisms are considered in the mechanistic model. *Biofilm-dependant speed*. A target speed is defined accordingly to the local density of biofilm and asymptotically reached after a relaxation time. *Biofilm-dependent direction*. Swimming direction is defined accordingly to the local biofilm density gradient. *Random walk*. A Brownian motion is added. (**d**) The image acquisition workflow is composed of a first step at the wet lab where host biofilm and swimmer are plated and imaged in different color channels. Then a post-processing phase recomposes the swimmer trajectories with tracking algorithms. Finally, temporal positions, speeds and accelerations are computed. On the biofilm channel, density and density gradient maps are processed at each time step.

for biofilm biocontrol, including spermatozoa (*Mayorga-Martinez et al., 2021*), protozoans (*Derlon et al., 2012*), or metazoans (*Klein et al., 2016*). Recent results suggest a deeper role of bacterial swimmers in biofilm ecology with the concept of microbial hitchhiking: motile bacteria can transport sessile entities such as spores (*Muok et al., 2021*), phages (*Yu et al., 2021*) or even other bacteria (*Samad et al., 2017*), enhancing their dispersion within the biofilm. Hence, characterizing microbial swimming in the very specific environment of the biofilm matrix is of particular interest to decipher

biofilm spatial regulations and their biocontrol, but more generally in an ecological perspective of microbial population dynamics in natural ecosystems.

Bacterial swimming is strongly influenced by the micro-topography and bacteria deploy strategies to sense and adapt their motion to their environment (*Lee et al., 2021*), with specific implications for biofilm formation and dynamics (*Conrad and Poling-Skutvik, 2018*). Model-based studies were conducted to characterize bacterial active motion in interaction with an heterogeneous environment. An image and model-based analysis showed non-linear self-similar trajectories during chemotactic motion with obstacles (*Koorehdavoudi et al., 2017*). Theoretical studies explored Brownian dynamics of self-propelled particles in interaction with filamentous structures such as EPS (*Jabbarzadeh et al., 2014*) or with random obstacles, exhibiting continuous limits and different motion regimes depending on obstacle densities (*Chepizhko and Peruani, 2013*; *Chepizhko et al., 2013*). Image analysis characterized different swimming patterns in polymeric fluids (*Patteson et al., 2015*), completed by detailed comparisons between a micro-scale model of flagellated bacteria in polymeric fluids and high-throughput images (*Martinez et al., 2014*). Models of bacterial swimmers in visco-elastic fluids were also developed to study the force fields encountered during their run (*Li and Ardekani, 2016*). However, to our knowledge, no study tried to characterize swimming patterns in the highly heterogeneous environment presented by an exogenous biofilm matrix.

In this study, we aim to provide a quantitative characterization of the different swimming behaviours in adaptation to the host biofilm matrix observed by microscopy. We focus on identifying potential species-dependent swimming characteristics and quantifying the swimming speed and direction variations induced by the host biofilm structure. To address these goals, three different *Bacillus* species presenting contrasted physiological characteristics are selected. First, different trajectory descriptors accounting for interactions with the host biofilm are defined, allowing to discriminate the swim of these bacterial strains by differential analysis. Then, a mechanistic random-walk model including swimming adaptations to the host biofilm is introduced. This model is numerically explored to identify the sensitivity of the trajectory descriptors to the model parameters. An inference strategy is designed to fit the model to 2D+T microscopy images. The method is validated on synthetic data and applied to a microscopy dataset to decipher the swimming behaviour of the three *Bacillus*.

## Results

### Ultrastuctural bacterial morphology

To investigate how the shape and propelling mechanism of bacteria can affect the way they navigate in a porous media such as a biofilm, we first image three bacterial swimmers –*Bacillus pumilus* (*B. pumilus*), *Bacillus sphaericus* (*B. sphaericus*), and *Bacillus cereus* (*B. cereus*) – by Transmitted Electron Microscopy (TEM) (*Figure 2*) to seek for potential structural and physiological differences. Important discrepancies can be observed between these *Bacillus*. First, they show noticeable difference in length and diameter, *B. sphaericus* being the longest bacteria by a factor of approximatively 1.5, and *B. cereus* and *B. pumilus* having similar size, but *B. cereus* showing a higher aspect ratio. Secondly, they do not have the same type of flagella: *B. pumilus* and *B. sphaericus* present several long flagella distributed over the whole surface of the membrane while *B. cereus* shows a unique brush-like bundle of very thin flagella, at its back tip.

We then used these three species to test if these ultrastructural differences could impact their swimming behaviour in a host biofilm or in a Newtonian control fluid: could the longer body of *B. sphaericus* be an impediment in a crowded environment such as a biofilm or on the contrary could its larger size give it a higher strength to cross the biofilm matrix? Is the unique brush-like flagella of *B. cereus* an advantage or a disadvantage to swim in a Newtonian fluid or in a host biofilm?

### Characterizing bacterial swimming in a biofilm matrix through image descriptors

2D+T Confocal Laser Scanning Microscopy (CLSM) of the three *Bacillus* swimming in a *Staphylococcus aureus* (*S. aureus*) host biofilm or in a control Newtonian buffer are acquired (see *Figure 1d*). Swimmers and host biofilms are imaged with different fluorescent dyes, allowing their acquisition in different color channels, and to recover in the same spatio-temporal referential the swimmer trajectories and the host biofilm density (see Materials and methods, *Figure 1* and *Table 1*). Namely, for each

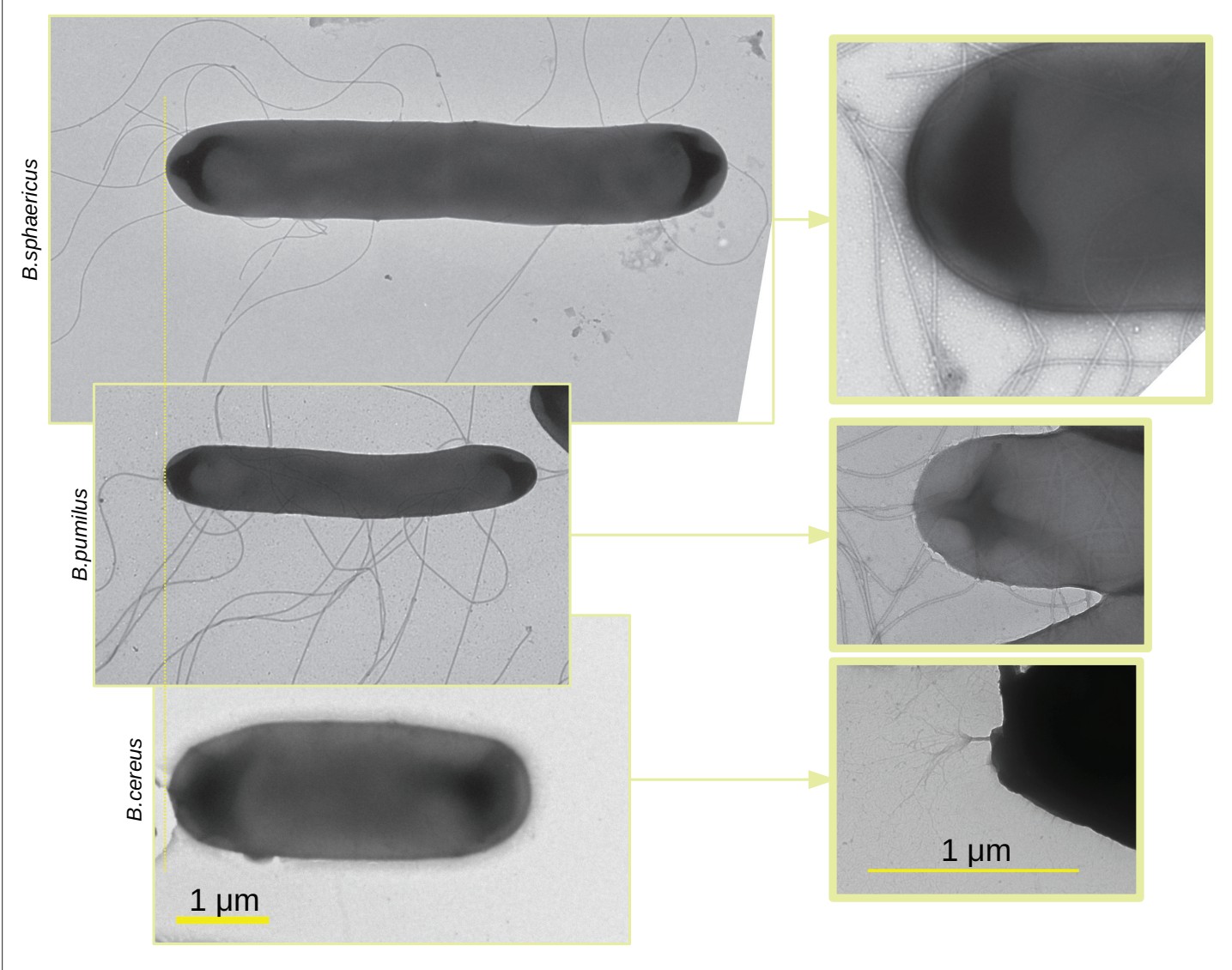

**Figure 2.** TEM images of the three *Bacillus*. TEM images of the three *Bacillus* are acquired, scaled in the same dimension and aligned (left panel). Images at lower scale are made with a zoom in on the flagella insertion (right panel). Note that the zoom in is optical so that the zoomed in image do not correspond to a zone of the larger scale images.

species $s$ and individual swimmer , we recover the initial ($T^s_{0,i}$) and final ($T^s_{end,i}$) observation times (when the swimmer goes in and out the focal plane, see Materials and methods sect. Confocal Laser Scanning Microscopy [CLSM]), and the number $T^s_i$ of time points in the trajectory. We then extract from the 2D+T images the observed position, instantaneous speed and acceleration time-series

$$t \mapsto X^s_i(t), \quad t \mapsto V^s_i(t), \quad t \mapsto A^s_i(t), \quad \text{for } t \in (T^s_{0,i}, T^s_{end,i}).$$

Noting $b^s(t, x)$ the dynamic biofilm density maps obtained from the biofilm images, we also compute the local biofilm density and density gradient along trajectories

$$t \mapsto b^s(t, X^s_i(t)), \quad \text{and } t \mapsto \nabla b^s(t, X^s_i(t)).$$

The angle $\theta^s_i(t)$ and the average velocity $\bar{V}^s_i(t)$ between two successive speed vectors are also collected (see Materials and methods sec. Post-processing of image data).

Different swimming patterns can be deciphered by qualitative observations of the trajectories $X^s_i(t)$ (*Figure 3*) in the biofilm and in the control Newtonian buffer, and run-and-tumble swimming patterns

are quantified with $\theta_i^s(t)$ and $\bar{V}_i^s(t)$ (**Figure 4**). *B. sphaericus* has a similar run-and-reverse behaviour in the biofilm and the control buffer with trajectories divided between back and forth paths around the starting point and long runs, the biofilm strongly impairing its speed and increasing the number of reverse events. By contrast, *B. pumilus* clearly switches its swimming behaviour in the biofilm, from quasi-straight runs in the Newtonian buffer to a pronounced run-and-reverse behaviour in the biofilm with decreased speeds and chaotic trajectories. On the contrary, *B. cereus* swimmers manage to conserve comparable trajectories and distributions of swimming speed and direction in the biofilm compared to control. Interestingly, the number of reverse events is even reduced in the host biofilm for *B. cereus*.

For further quantitative analysis, trajectory descriptors are defined. We first investigate the distribution of the population-wide average acceleration and velocity norms $\frac{1}{T_i^s-2}\sum_t \|A_i^s(t)\|$ and $\frac{1}{T_i^s-1}\sum_t \|V_i^s(t)\|$, where $\|\cdot\|$ denotes the Euclidian norm. We also quantify the swimming kinematics by computing the travelled distance $dist_i^s$ along the path and the total displacement $disp_i^s$, that is the distance between the initial and final trajectory points, with

$$dist_i^s = \int_{T_{0,i}^s}^{T_{end,i}^s} \|V_i^s(t)\|dt \quad \text{and} \quad disp_i^s = \|X(T_{end,i}^s) - X(T_{0,i}^s)\| = \|\int_{T_{0,i}^s}^{T_{end,i}^s} V_i^s(t)dt\|.$$

We finally compute the total biofilm area visited by a swimmer along its path (see **Figure 1b**). The same descriptors are computed in the control Newtonian buffer.

The three species present contrasted distributions for these descriptors (**Figure 5**). *B. sphaericus* has the smallest mean ($\|A\| = 0.58$ and $\|V\| = 0.70$) and median ($\|A\| = 0.50$ and $\|V\| = 0.53$) values of acceleration and speed, while *B. pumilus* has the widest distributions (difference between 95% and 5% centiles of 2.76 for $\|A\|$ and 2.45 for $\|V\|$ compared to 1.00, 1.51 and 1.90, 1.49 for *B. sphaericus* and *B. cereus* respectively). *B. cereus* for its part shows the highest accelerations, indicating larger changes in swimming velocities, but median and mean speeds comparable to *B. pumilus* (**Figure 5**, $\|A\|$ and $\|V\|$ panels). We also note that *B. sphaericus* and to a lower extent *B. pumilus* trajectories have a significant amount of null or small average speeds, while *B. cereus* trajectories have practically no zero velocity, consistently with the qualitative analysis (**Figure 5**, $\|V\|$ panels). Small velocities episodes of *B. sphaericus* and *B. pumilus* could occur during their back-and-forth trajectories, which produce small displacements and pull the displacement distribution towards lower values than *B. cereus* (**Figure 5**, *Disp* panel). *B. pumilus* displacement is intermediary. Conversely, back-and-forth trajectories can produce large swimming distances for *B. sphaericus* and *B. pumilus* (mean adimensioned value of 32.2 and 43.2 respectively) so that *B. sphaericus* has a distance distribution comparable to *B. cereus* (mean adimensioned value of 29.6, **Figure 5**, *Dist* panel), but lower than *B. pumilus*. Observing conjointly displacement and distance (**Figure 5**, lower-right panel) provides consistent insights: *B. sphaericus* shows a large variability of small displacement trajectories, from small to large distances,

**Table 1.** Dataset characteristics.

We detailed, for each batch, the number of trajectories, the average number of time points by trajectory (and standard deviation), the total number of time points in the dataset, the total movie duration in seconds and the time interval between two snapshots in seconds.

| Species | Batch | # traject. | traj. length | time points | Duration [s] | $\Delta t$[s] |
|---|---|---|---|---|---|---|
| *B. pumilus* | 1 | 122 | 40 (7.4) | 4,590 | 30 | 0.134 |
| | 2 | 152 | 25 (5.7) | 3,543 | 30 | 0.134 |
| | 3 | 243 | 38 (6.9) | 8,825 | 30 | 0.134 |
| *B. sphaericus* | 1 | 98 | 40 (7.6) | 3,762 | 30 | 0.134 |
| | 2 | 91 | 43 (7.7) | 3,771 | 30 | 0.134 |
| | 3 | 48 | 55 (7.9) | 2,543 | 23 | 0.134 |
| *B. cereus* | 1 | 105 | 47 (7.9) | 4,766 | 30 | 0.069 |
| | 2 | 53 | 36 (7.7) | 1,808 | 30 | 0.069 |
| | 3 | 121 | 43 (7.1) | 5,006 | 30 | 0.069 |

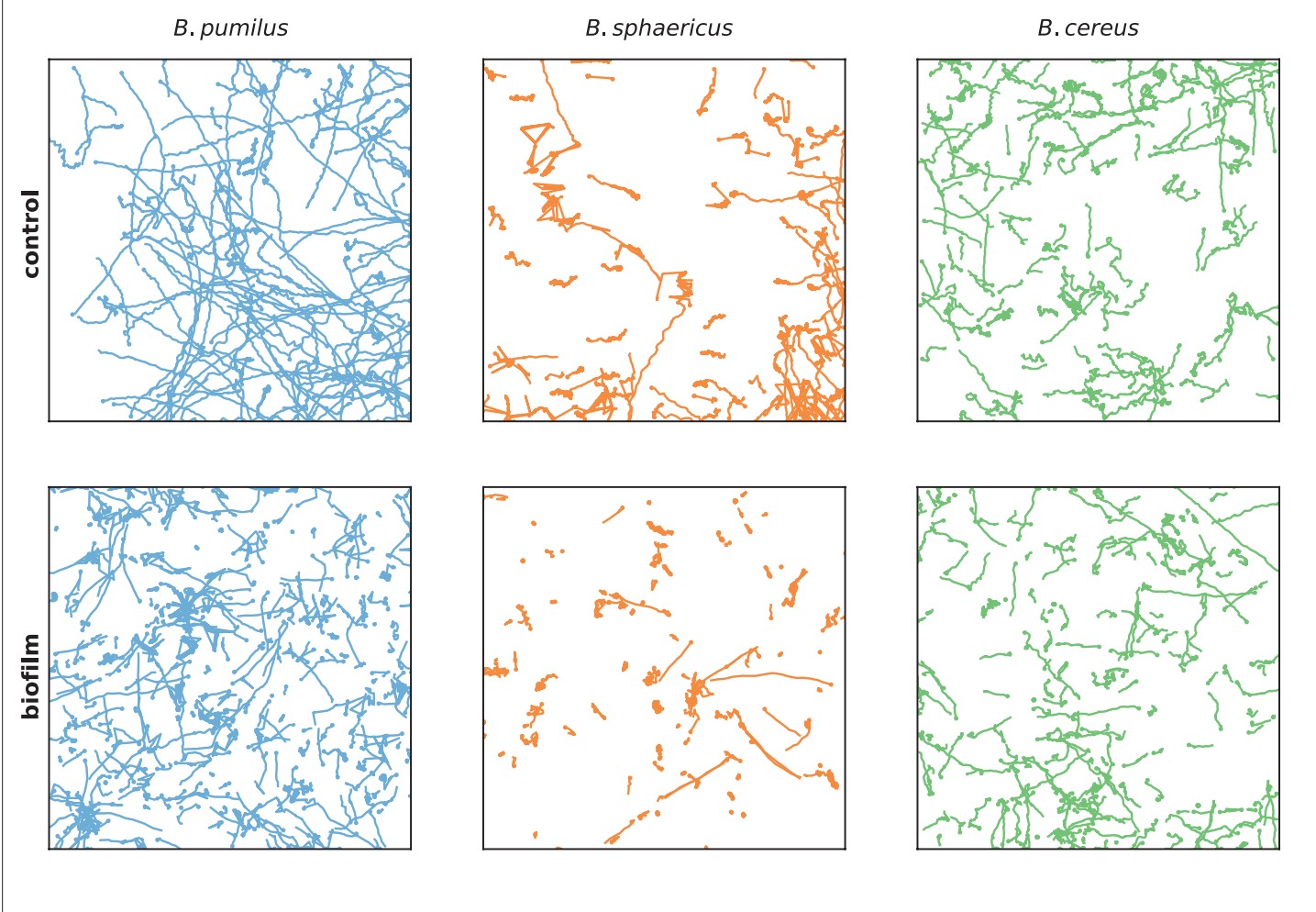

**Figure 3.** Swimmer trajectories The whole set of trajectories of each species is displayed in the control Newtonian buffer (upper panel) and in the host biofilm (lower panel). Note that the 3 batches of the different species are pooled on these images. Number of trajectories are n = 517 and 123 (*B. pumilus*), n = 237 and 94 (*B. sphaericus*) and n = 279 and 144 (*B. cereus*) for, respectively, the biofilm and the control buffer. The physical size of the domain is 147x147μm.

while *B. cereus* trajectory displacement seems to vary almost linearly with the distance at least for the points inside the isoline 50%. *B. pumilus* has again an intermediary distribution, with a large range of displacement-distance couples. The distributions of visited areas of *B. pumilus* and *B. cereus* are almost identical, and higher than *B. sphaericus* one. Compared to the control buffer, all descriptors are reduced in the biofilm. Consistently with previous observations, the displacement (*disp*) is strongly reduced for *B. pumilus*, and less impacted for *B. sphaericus* and *B. cereus*. These observations must be related to the behavioural switch for *B. pumilus* and to the identical swimming patterns for the two other *Bacilii* in the biofilm compared to the control fluid.

All together, this data depict (1) a long-range species, *B. cereus*, which moves efficiently in the biofilm during long, relatively straight, rapid runs, almost identically as in a Newtonian fluid (2) a short-range species, *B. sphaericus*, that moves mainly locally in small areas in the biofilm and in the control buffer with lower accelerations and speeds except few exceptions (only 6% of its trajectories induced a displacement higher than $10\mu m$ compared to 28% for *B. cereus* and 26% for *B. pumilus*) and (3) a medium-range species, *B. pumilus*, with a large diversity of rapid trajectories, from small to large displacement, and a behavioural change from straight runs in a Newtonian fluid to frequent run-and-reverse events in the biofilm. These kinematics discrepancies for *B. pumilus* and *B. cereus* allow them however to cover identical visited areas.

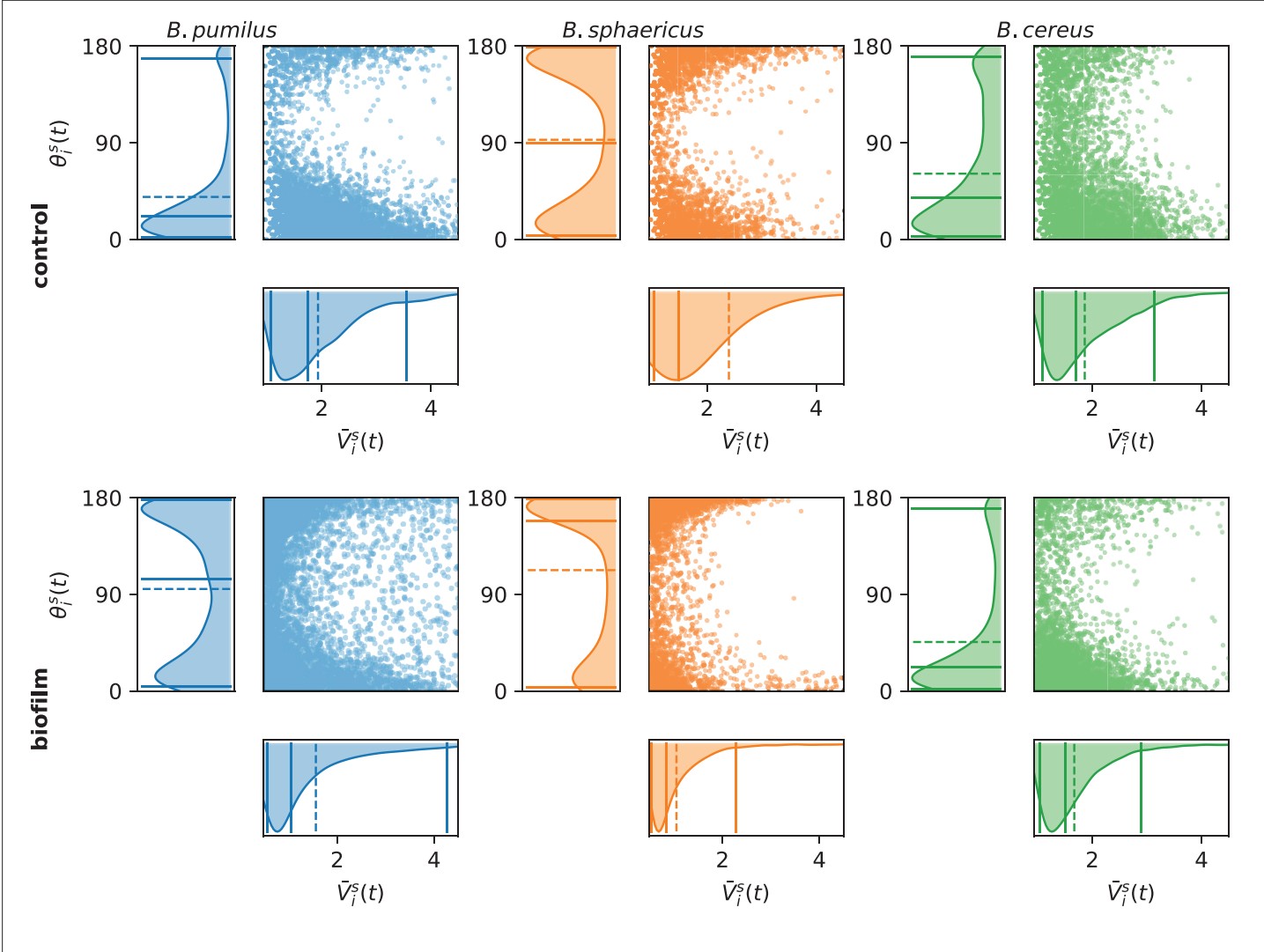

**Figure 4.** Assessing run-and-tumble with speed and direction distributions. For each time point, the swimmer mean speed $\bar{V}_i^s(t)$, defined as the mean between the incoming and outgoing velocity vectors $\bar{V}_i^s(t) = (\|V_i^s(t)\| + \|V_i^s(t - \Delta t)\|)/2$, for $t \in (T_{0,i}^s + \Delta t, T_{end,i}^s)$, is plotted versus the direction change, defined as the angle $\theta_i^s(t)$ between the incoming and outgoing velocity vectors $\theta_i^s(t) = \arccos((V_i^s(t) \cdot V_i^s(t - \Delta t))/(\|V_i^s(t)\|\|V_i^s(t - \Delta t)\|))$. The left and bottom panels indicate the marginal distributions, with the mean (dashed line) and quantiles 0.05, 0.5, and 0.95 (plain lines). Number of times points are n = 6 848 and 6 509 (*B.pumilus*), n = 2 818 and 3 740 (*B. sphaericus*) and n = 3 526 and 4 435 (*B. cereus*) for, respectively, the biofilm and the control buffer.

Though, these global descriptors do not inform about potential adaptations of the swimmers to the biofilm matrix. We first check if swimmer velocities are directly linked to the local biofilm density, and if the swimmers adapt their trajectory according to density gradients by plotting the points $(\|\nabla b(t, X_i^s(t))\|, \|A_i^s(t)\|)$ and $(b(t, X_i^s(t)), \|V_i^s(t)\|)$ (*Figure 5*, lower panel). Clear differences between the three species can be deciphered. First, the three *Bacillus* do not have the same distribution of visited biofilm density and gradient. *B. pumilus* swimmers visit denser biofilm with higher variations than the other species while *B. sphaericus* and *B. cereus* stay in less dense and smoother areas, the quantile 0.5 of these species being circumscribed in low gradient and low density values. Next, *B. cereus* has a wider distribution of accelerations, specially for small-density gradients, compared to *B. pumilus* and *B. sphaericus*. This could indicate that when the biofilm is smooth, *B. cereus* samples its acceleration in a large distribution of possible values. Finally, we observe that the speed distribution rapidly drops for increasing biofilm densities for *B. sphaericus* and *B. cereus*, while the decrease is much smoother for *B. pumilus*. These observations provide additional insights in the species swimming characteristics: *B. pumilus* swimmers seem to be less inconvenienced by the host biofilm density than the other species, while *B. cereus* and *B. sphaericus* bacteria appear to be particularly impacted by higher densities and

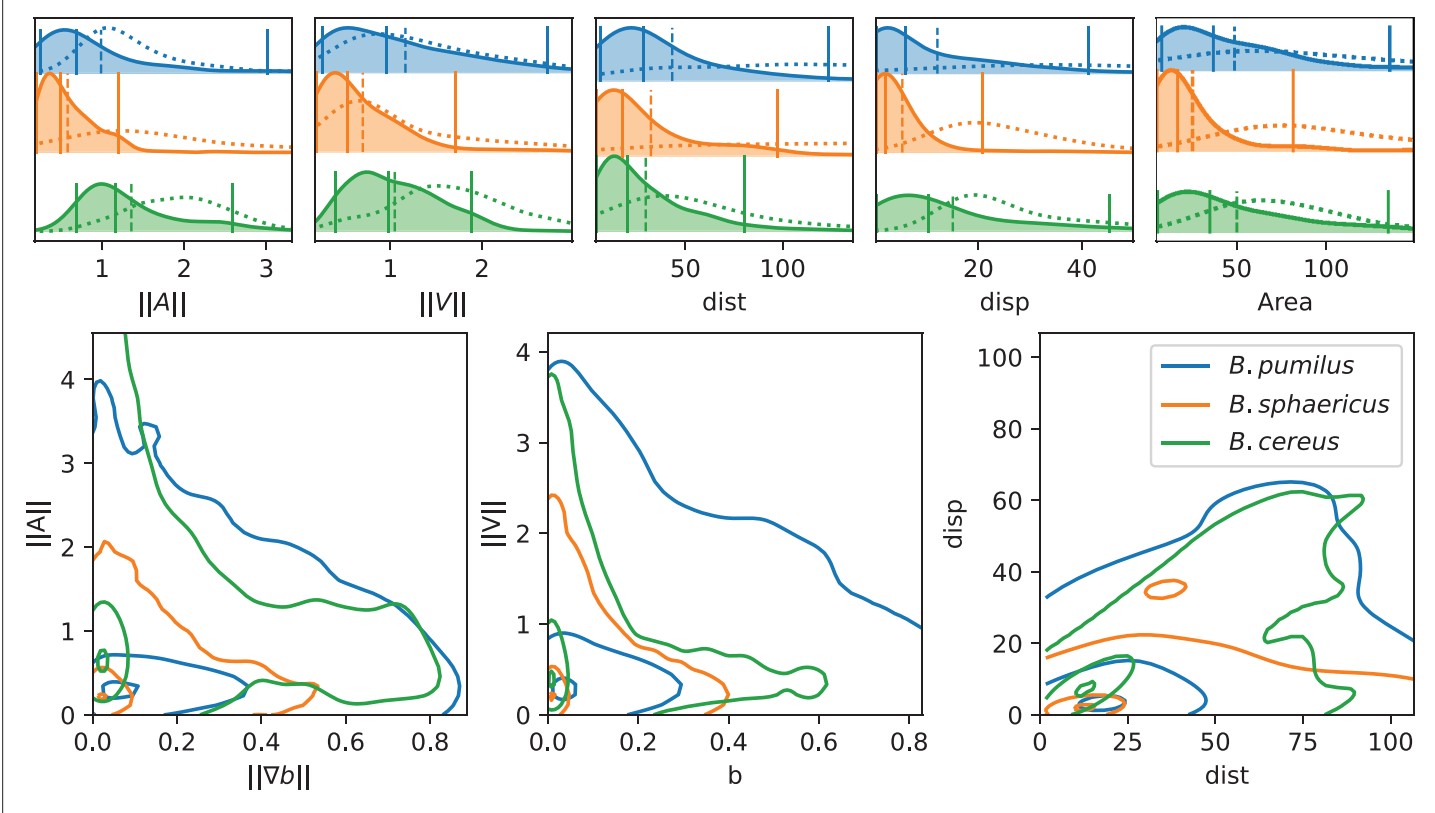

**Figure 5.** Analysis of swimming characteristics using trajectory descriptors. Upper panel: normalized acceleration, speed, distance, displacement, and area distributions structured by species are displayed, together with quantile 0.05, 0.5, and 0.95 (vertical plain lines) and mean (vertical dashed line). The descriptor distribution in the control Newtonian buffer is indicated with the dotted line. All values are normalized by the corresponding reference value as indicated in Materials and methods. T-test pairwise comparison p-values are displayed in *Appendix 1—table 2*. Number of trajectories are n = 517 and 123 (*B.pumilus*), n = 237 and 94 (*B. sphaericus*) and n = 279 and 144 (*B. cereus*) for, respectively, the biofilm and the control buffer. Lower panel: we display the distribution of the instantaneous acceleration norm respectively to the local biofilm density gradient (i.e. ‖$A_i(t)$‖ function of ∇$b(X_i(t))$) and of the instantaneous velocity norm respectively to the local biofilm density (i.e. ‖$V_i(t)$‖ function of $b(X_i(t))$), structured by population. The point cloud of each species is approximated by a gaussian kernel and gaussian kernel isolines enclosing 5, 50% and 95% of the points centered in the densest zones are displayed to facilitate comparisons between species (see Materials and methods Plots and statistics).

to favor low densities where it can efficiently move. Though, *B. sphaericus* has lower motile capabilities than *B. cereus* when the biofilm is not dense.

## Analysis of swimming data with an integrative swimming model

This descriptive analysis does not allow to clearly identify potential mechanisms by which the swimmers adapt their swim to the biofilm structure or to simulate new species-dependant trajectories. We then build a swimming model based on a Langevin-like equation on the acceleration that involves several swimming behaviours modelling the swimmer adaptation to the biofilm. Furthermore, after inference, new synthetic data can be produced by predicting swimmer random walks sharing characteristics comparable to the original data.

We consider bacterial swimmers as Lagrangian particles and we model the different forces involved in the update of their velocity **v**. We assume that the swimmer motion can be modelled by a stochastic process with a deterministic drift (*Figure 1c*):

$$d\mathbf{v} = \underbrace{\gamma(\alpha(b) - \|\mathbf{v}\|)\frac{\mathbf{v}}{\|\mathbf{v}\|}dt}_{\text{speed selection}} + \underbrace{\beta\frac{\nabla b}{\|\nabla b\|}dt}_{\text{direction selection}} + \underbrace{\eta dt}_{\text{random term}} \qquad (1)$$

where the right hand side is composed of two deterministic terms in addition to a gaussian noise, each weighted by the parameters $\gamma$, $\beta$ and $\epsilon$.

The first term implements the biological observation (**Figure 5**, lower central panel) that the bacterial swimmers adapt their velocity to the biofilm density. This term can be interpreted as a speed selection term that pulls the instantaneous speed of the swimmer towards a prescribed target velocity $\alpha(b)$ that depends on the host biofilm density $b$. The weight $\gamma$ can be interpreted as a penalization coefficient. In such a formalism, the difference between the swimmer and the prescribed speed is divided by a relaxation time $\tau$ to be homogeneous to an acceleration. Hence, $\gamma$ is proportionally inverse to $\tau$, $\gamma \sim \frac{1}{\tau}$. As a first-order approximation of the speed drop observed in **Figure 5** for increasing $b$, the target speed $\alpha(b)$ is modeled as a linear variation between $v_0$ and $v_1$, where $v_0$ is the swimmer characteristic speed in the lowest density regions, where $b = 0$, and $v_1$ in the highest density zones where $b = 1$:

$$\alpha(b) = v_0(1 - b) + bv_1 = v_0 + b(v_1 - v_0)$$

The second term updates the velocity direction according to the local gradient of the biofilm density $\nabla b$. The sign of $\beta$ indicates if the swimmer is inclined to go up (negative $\beta$) or down (positive $\beta$) the host biofilm gradient, while the weight magnitude indicate the influence of this mechanism in the swimmer kinematics. We note that this term does not depend on the gradient magnitude but only on the gradient direction: this reflects the implicit assumption that the bacteria are able to sense density variations to find favorable directions, but that the biological sensors are not sensitive enough to evaluate the variation magnitudes.

The third term is a stochastic two-dimensional diffusive process that models the dispersion around the deterministic drift modelled by the two first terms. We define

$$\eta \sim \mathcal{N}(0, \epsilon)$$

The term $\eta$ can also be interpreted as a model of the modelling errors, tuned by the term $\epsilon$. **Equation 1** is supplemented by an initial condition by swimmer. For vanishing $\|v\|$ or $\|\nabla b\|$ leading to an indetermination, the corresponding term in the equation is turned off.

**Equation 1** links the observed biofilm density and the swimmer trajectories trough mechanistic swimming behaviours. The model fitting can be seen as an ANOVA-like integrative statistical analysis of the image data. It decomposes the observed acceleration variance between mechanistic processes describing different swimming traits in order to decipher their respective influence on the swimmer trajectories while integrating heterogeneous data (density maps $b$ and trajectories kinematics).

We can define characteristic speed and acceleration $V^*$ and $A^*$ in order to set a dimensionless version of **Equation 1**

$$d\mathbf{v} = \gamma'(v_0' + b(v_1' - v_0') - \|\mathbf{v}\|)\frac{\mathbf{v}}{\|\mathbf{v}\|}dt + \beta'\frac{\nabla b}{\|\nabla b\|}dt + \eta'dt \tag{2}$$

where $\gamma' = \frac{\gamma V^*}{A^*}$, $v_0' = \frac{v_0}{V^*}$, $v_1' = \frac{v_1}{V^*}$, $\beta' = \frac{\beta}{A^*}$, $\eta' \sim \mathcal{N}(0, \epsilon')$ and $\epsilon' = \frac{\epsilon}{A^{*2}}$.

This dimensionless version will strongly improve the inference process and will allow an analysis of the relative contribution of the different terms in the kinematics. An extended numerical exploration of this model is performed in Appendix 2 Sec. Numerical exploration on mock biofilm images to illustrate the impact of the different parameters on the trajectories, showing in particular the interplay between $\gamma$ and $\epsilon$: counter-intuitively, straight lines are induced when the stochastic part $\epsilon$ is high compared to the speed selection parameter $\gamma$ (see also Appendix 2).

## Inferring swimming parameters from trajectory data

For each bacterial swimmer population, we now seek to infer with a Bayesian method population-wide model parameters governing the swimming model of a given species from microscope observations.

### Inference model setting

**Equation (2)** is re-written as a state equation on the acceleration for the bacterial strain $s$ and the swimmer

$$A_i^s(t) = \gamma^s(v_0^s + b(t, X_i^s(t))(v_1^s - v_0^s) - \|V_i^s(t)\|)\frac{V_i^s(t)}{\|V_i^s(t)\|} + \beta^s\frac{\nabla b(t, X_i^s(t))}{\|\nabla b(t, X_i^s(t))\|} + \eta^{\mathbf{s}} \tag{3}$$

$$:= f_A\left(\theta^s, b(t, X_i^s(t)), V_i^s(t), X_i^s(t)\right) + \eta^{\mathbf{s}} \tag{4}$$

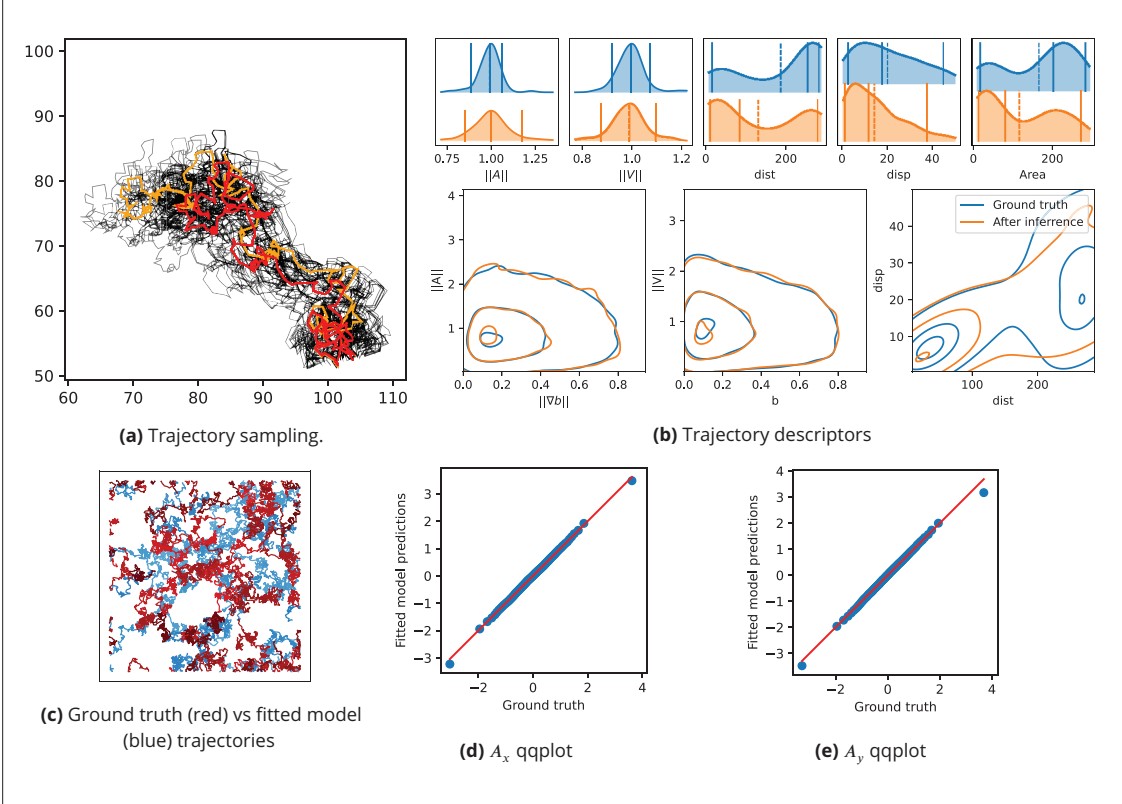

**(a)** Trajectory sampling.

**(b)** Trajectory descriptors

**(c)** Ground truth (red) vs fitted model (blue) trajectories

**(d)** $A_x$ qqplot

**(e)** $A_y$ qqplot

**Figure 6.** Inference assessment on synthetic data. (**a**) Predicted vs true trajectories. Trajectories are recovered by sampling the parameter posterior distribution starting from the same initial condition than in the data. We represent a ground truth trajectory extracted randomly from the original dataset in red, the corresponding sampled trajectories with thin gray lines, and the trajectory obtained with the posterior means in orange. Note that in this simulation, the stochastic part is the same for all simulations, so that the only source of uncertainties comes from the inference procedure. (**b**) Trajectory descriptors. Trajectories are re-computed replacing the original parameters (ground truth) by the inferred parameters. The trajectory descriptors introduced in Characterizing bacterial swimming in a biofilm matrix through image descriptors are computed on the synthetic data (blue curves) and on the data obtained with the inferred parameters (orange curves). Number of trajectories are n = 72 for the ground truth and n = 100 after inference. (**c**) Ground truth vs fitted trajectories. The ground truth, that is the original trajectories (blue) and fitted (red) trajectories are displayed and show common characteristics. (**d-e**) Qqplot of fitted model output vs ground truth. After inference, the fitted model is used to re-compute the synthetic dataset. We plot the x (**d**) and y (**e**) components of the accelerations in a qqplot: the fitted model output quantiles are plotted against the quantiles of the original dataset (ground truth) with blue dots, together with the $y = x$ line (red).

where

$$\theta^s := (\gamma^s, v_0^s, v_1^s, \beta^s)$$

are species-dependant equation parameters. The function $f_A$ can be seen as the deterministic drift of the random walk, gathering all the mechanisms included in the model. The inter-individual variability of the swimmers of a same species comes from the swimmer-dependent initial condition, the resulting biofilm matrix they encounter during their run, and the stochastic term.

Inferring the parameters $\theta^s$ can then be stated in a Bayesian framework as solving the non linear regression problem

$$A_i^s(t) \sim \mathcal{N}\left(f_A\left(\theta^s | b(t, X_i^s(t)), V_i^s(t), X_i^s(t)\right), \epsilon^s\right) \tag{5}$$

from the data $b(t, X)$, $X_i^s(t)$, $V_i^s(t)$ and $A_i^s(t)$, with truncated normal prior distributions

$$\theta^s \sim \mathcal{N}(0, 1), \quad \epsilon^s \sim \mathcal{N}(0, 1), \tag{6}$$

and additional constrains on the parameters

$$\gamma^s \geq 0, \quad v_0^s \geq 0, \quad v_1^s \geq 0, \quad \epsilon^s \geq 0.$$

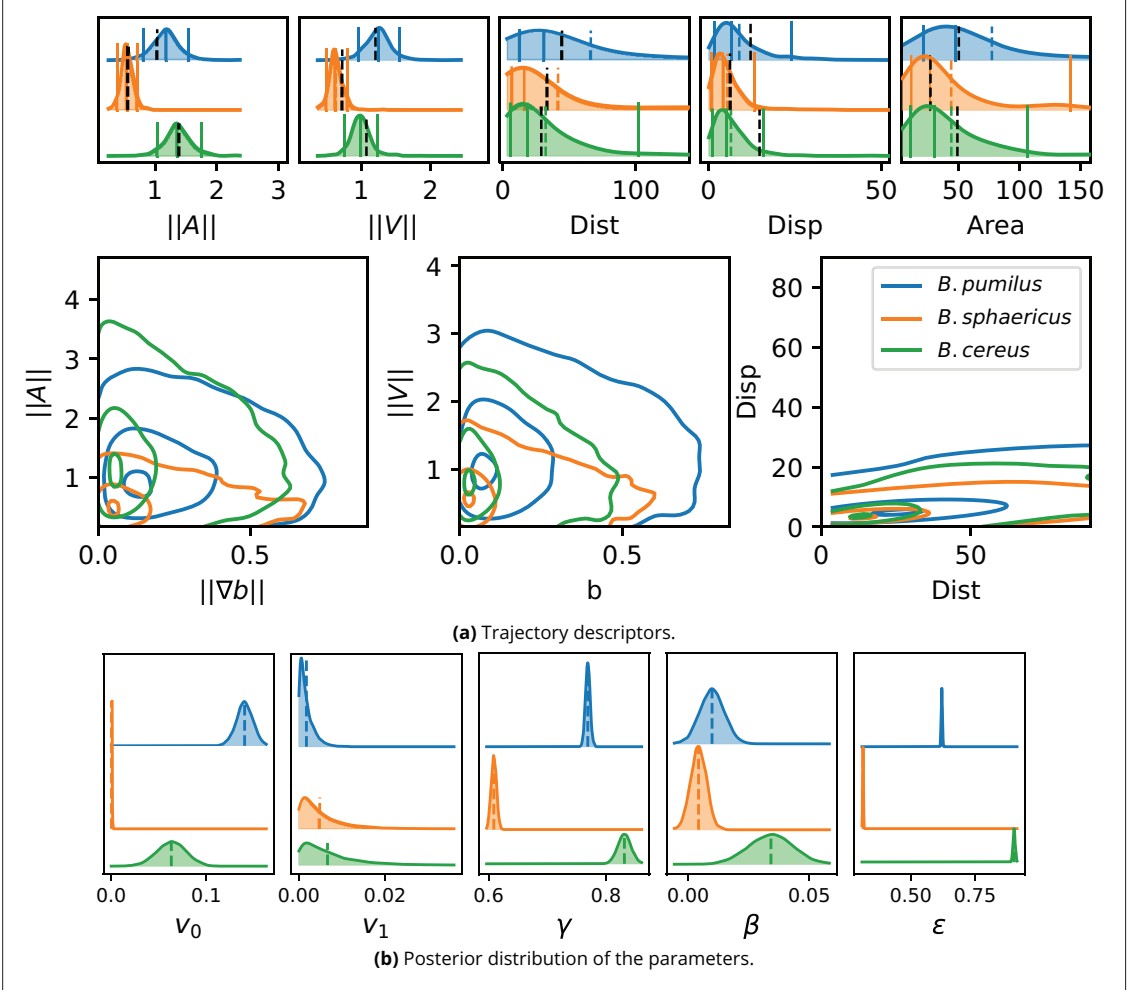

**(a)** Trajectory descriptors.

**(b)** Posterior distribution of the parameters.

**Figure 7.** Inference result on the experimental images. (**a**) To validate the inference process, a synthetic dataset is assembled by computing *Equation 1* with the inferred parameters and the trajectory descriptors introduced in section Characterizing bacterial swimming in a biofilm matrix through image descriptors are computed and can be compared to the data descriptors in *Figure 5*. Acceleration, speed, distance and displacement distributions are displayed in the upper panel, with quantiles 0.05, 0.5 and 0.95 (plain lines) and mean (dashed line). The mean values observed in the image data are also displayed for comparison (black dashed line). The number of trajectories are identical than in *Figure 5*: n = 517 (*B. pumilus*), n = 237 (*B. sphaericus*) and n = 279 (*B. cereus*). Interactions between the host biofilm and, respectively, acceleration and speed distributions are displayed in the lower panel with isolines enclosing 5, 50% and 95% of the points, centered in the densest zones. (**b**) Inferred parameter posterior distributions after analysis of the confocal swimmer images, and posterior mean (dashed line). We used 4000 points for the computation of the Gaussian KDE.

We note that *Equation (5)* can be seen as a likelihood equation of the parameter $\theta^s$ knowing $A_i^s(t), b(t), V_i^s(t)$ and $X_i^s(t)$. The parameter $\epsilon^s$ can now be seen as a corrector of both modelling errors in the deterministic drift and observation errors between the observed and the true instantaneous acceleration. Alternative settings where these uncertainties sources are separated and a true state for position and acceleration is inferred can be defined (see Annex Various inference models). The inference problem is implemented in the Bayesian HMC solver Stan (*Stan Development Team, 2018*) using its *python* interface *pystan* (*Riddell et al., 2021*). Inference accuracy is thoroughly assessed on synthetic data (see Appendix 1 Assessment of the inference with synthetic data and *Figure 6*).

## Analysis of the confocal microscopy dataset

We now solve the inference problem (5)-(6) on the confocal microscopy dataset to identify population-wide swimming model parameters in order to decompose the swimmer kinematics in three mechanisms: biofilm-related speed selection, density-induced direction changes and random walk. The inference process is assessed by comparing the descriptors obtained on trajectories predicted by the

fitted model (*Figure 7a*) with descriptors of real trajectories (*Figure 5*). The mean values of acceleration and speeds are accurately predicted for the three species ( *Figure 7a* panels $\|A\|$ and $\|V\|$, dashed lines). Relative positions of distance, displacement and visited area mean values are also correctly simulated (*Figure 5 and Figure 7a*, upper panel). *B. sphaericus* presents the lowest predicted accelerations and speeds while *B. pumilus* has the widest speed and acceleration distributions and *B. cereus* shows the highest accelerations, consistently with the data. The visited area and the distances are slightly over estimated, but the relative position and the shape of the distributions are conserved. The amount of null velocities for *B. sphaericus* is under estimated by the fitted model and not rendered for *B. pumilus*. The distance distributions of the three species are accurately predicted by the fitted model. When displaying conjointly the distance and the displacement (*Figure 7a*, right lower panel), the distribution of *B. sphaericus* is correctly predicted by the simulations, but *B. cereus* and *B. pumilus* displacements are underestimated. Some qualitative features can be recovered, such as the higher distribution of distance-distribution couples for *B. cereus* or higher displacement for *B. cereus* compared to *B. sphaericus*.

Descriptors of swimming adaptations to the host biofilm are also correctly preserved for the main part (*Figure 5 and Figure 7* a, lower panel). *B. pumilus* is the species that crosses the highest biofilm densities in the fitted model simulations, showing the highest speeds in this crowded areas, and that visits the most frequently areas with high density gradients, consistently with the data. As in the confocal images, the simulated *B. sphaericus* and *B. cereus* favor smoother zones of the biofilm with lower biofilm densities. The *B. cereus* fitted model correctly renders the highest acceleration variance observed in the data for low biofilm gradients, while *B. sphaericus* speed and acceleration variance is the lowest for all ranges of biofilm densities and gradients, both in the data and in the fitted model predictions. The drop of speeds and accelerations for increasing biofilm densities and gradients is well predicted for *B. pumilus*, but is smoother in the simulation compared to the data for *B. sphaericus* and *B. cereus*. In particular, the sharp drop of speeds for $b \simeq 0.25$ observed in the data for *B. cereus* and *B. sphaericus* is underestimated by the fitted model.

All together, the model reproduces very accurately the mean values of acceleration, speed and visited area, renders relative positions and the main characteristics of distributions for distance, displacement and interactions with the host biofilm matrix, but produces less variable outputs than observed in the data, meaning that the model is less accurate in the distribution tails. The main features of the swimmer adaptation to the underlying biofilm are however correctly predicted by the model.

To further inform the fitted model accuracy, the coefficient of determination $R^2_{det}$ of the deterministic components $f_A(\theta^s, b(t), V^s_i, X^s_i(t))$ of *Equation 4* is computed (*Table 2*), in order to quantify the goodness of fit of the friction and gradient terms of (*Equation 2*) that represent interactions with the

**Table 2.** Reference acceleration and speed, and acceleration variance decomposition between stochastic and deterministic terms.

The number $N$ of acceleration time points is indicated for each specie. Then, reference values for acceleration $A_{ref}$ and speed $V_{ref}$ used for adimensionalization are computed by averaging the corresponding values by specie. Descriptive statistics of acceleration variance decomposition are then computed in order to illustrate the contribution of the deterministic terms in the observed acceleration distribution, and the part of the residual mechanisms that are not included in the model. We indicate for each species the acceleration variance $\sigma(A)$, the part of the variance explained by the deterministic terms $R^2_{det}$ (see Materials and methods sec.Inference validation on experimental data) and the variance of the stochastic term $\epsilon^2$. We note that in order to compare species at vizualisation step, they are re-normalized with the average of the species reference values: $A_{ref} = 78.31$ and $V_{ref} = 6.55$.

| data | N | $A_{ref}$ | $V_{ref}$ | $\sigma(A)$ | $R^2_{det}[\%]$ | $\epsilon^2$ |
|---|---|---|---|---|---|---|
| B.pumilus | 33,916 | 81.08 | 7.89 | 0.87 | 58.80 | 0.36 |
| B.sphaericus | 20,152 | 44.93 | 4.74 | 0.58 | 48.50 | 0.30 |
| B.cereus | 23,160 | 108.92 | 7.03 | 0.63 | 32.72 | 0.42 |

**Table 3.** Inference outputs for the three species.
The posterior mean, standard deviation and inferred confidence interval are indicated for each parameter and each specie. Convergence diagnosis index $n_{eff}$ and $R_{hat}$ are provided.

| species | param | mean | std | confidence interval [2.5%–97.5%] | $n_{eff}$ | $R_{hat}$ |
|---|---|---|---|---|---|---|
| *B. pumilus* | $\gamma$ | 0.77 | $3.95 \times 10^{-3}$ | $[0.77–0.77]$ | 4,507 | 1 |
| | $v_0$ | 0.14 | $8.67 \times 10^{-3}$ | $[0.12–0.16]$ | 3,879 | 1 |
| | $v_1$ | $1.69 \times 10^{-3}$ | $1.69 \times 10^{-3}$ | $[5.18 \times 10^{-5} – 6.26 \times 10^{-3}]$ | 4,821 | 1 |
| | $\beta$ | $9.84 \times 10^{-3}$ | $5.07 \times 10^{-3}$ | $[1.45 \times 10^{-5} – 2.07 \times 10^{-2}]$ | 5,223 | 1 |
| | $\epsilon$ | 0.62 | $2.48 \times 10^{-3}$ | $[0.61–0.62]$ | 5,307 | 1 |
| *B. sphaericus* | $\gamma$ | 0.61 | $4.53 \times 10^{-3}$ | $[0.60–0.62]$ | 4,965 | 1 |
| | $v_0$ | $2.75 \times 10^{-4}$ | $2.75 \times 10^{-4}$ | $[4.91 \times 10^{-6} – 1.01 \times 10^{-3}]$ | 4,019 | 1 |
| | $v_1$ | $4.84 \times 10^{-3}$ | $4.77 \times 10^{-3}$ | $[9.39 \times 10^{-5} – 1.45 \times 10^{-2}]$ | 5,001 | 1 |
| | $\beta$ | $4.25 \times 10^{-3}$ | $3.33 \times 10^{-3}$ | $[-2.18 \times 10^{-3} – 1.15 \times 10^{-2}]$ | 4,668 | 1 |
| | $\epsilon$ | 0.32 | $1.55 \times 10^{-3}$ | $[0.31–0.32]$ | 5,943 | 1 |
| *B. cereus* | $\gamma$ | 0.83 | $1.11 \times 10^{-2}$ | $[0.80–0.86]$ | 2,700 | 1 |
| | $v_0$ | $6.44 \times 10^{-2}$ | $1.07 \times 10^{-2}$ | $[3.22 \times 10^{-2} – 9.66 \times 10^{-2}]$ | 2,510 | 1 |
| | $v_1$ | $6.65 \times 10^{-3}$ | $6.33 \times 10^{-3}$ | $[1.50 \times 10^{-4} – 2.15 \times 10^{-2}]$ | 4,061 | 1 |
| | $\beta$ | $2.78 \times 10^{-2}$ | $9.04 \times 10^{-3}$ | $[1.39 \times 10^{-2} – 5.56 \times 10^{-2}]$ | 4,230 | 1 |
| | $\epsilon$ | 0.9 | $4.17 \times 10^{-3}$ | $[0.89–0.92]$ | 4,852 | 1 |

biofilm. These results highlight that *B. cereus* bacteria do present an important stochastic part in the accelerations, while the *B. pumilus* species is the best represented by our deterministic modelling.

The three species present very different inferred parameter values (*Figure 7* b and *Table 3*), showing that the model inference captures contrasted swimming characteristics of these *Bacillus*. Due to the mechanistic terms introduced in *Equation 1*, these differences can be interpreted in term of speed and direction adaptations to the host biofilm. First, *B. pumilus* shows the highest $v_0$ value, and the highest amplitude between $v_0$ and $v_1$, inducing a higher ability for *B. pumilus* to swim fast in low density biofilm zones and strong deceleration in crowded area. In comparison, *B. sphaericus* presents the smallest amplitude between $v_0$ and $v_1$ showing a poor adaptation to biofilm density. *B. cereus* has the highest $\gamma$ value, showing a reduced relaxation time toward the density dependant speed: in other words, *B. cereus* is able to adapt its swimming speed more rapidly than the other species when the biofilm density varies. *B. cereus* swimmers are also better able to change their swimming direction in function of the biofilm variations they encounter along their way, their $\beta$ distribution being markedly higher than the other species which have very low $\beta$. Finally, the stochastic parameter $\epsilon$ is also contrasted, from a low distribution for *B. sphaericus* to high values for *B. cereus*. All together, the inference complete the observations made in *Figure 5*: *B. pumilus* poorly adapts its swimming direction to the host biofilm (low $\beta$) but has a wide range of possible speeds when the biofilm density varies (high $v_0$, low $v_1$), that it can reach quite rapidly (intermediary $\gamma$) with intermediary stochastic correction ($\epsilon$). In contrast, *B. cereus* reaches lower speed values (intermediary $v_0$, low $v_1$) but is more agile to adapt its swimming to its environment by changing rapidly its speed when the biofilm density is more favorable (highest $\gamma$) and adapting its swimming direction to biofilm variations, with higher stochastic variability (large $\epsilon$). Finally, *B. sphaericus* is the less flexible of the three bacteria: less fast (smallest difference between $v_0$ and $v_1$), they are also less responsive to biofilm variations (small $\gamma$ and $\beta$) with low random perturbations (small $\epsilon$).

Finally, after inference, the impact of each term in the overall acceleration data can be quantified and analyzed by displaying its relative contribution in a ternary plot (*Appendix 2—figure 6*). This relative contribution can be measured thanks to the swimming model which integrates these different mechanisms in the same inference problem. The direction selection is the least influential mechanism

for the three species, with a slightly higher impact for *B. cereus* (50% and 95% isolines slightly shifted towards $A(\nabla b)$ in *Appendix 2—figure 6a*). When zooming in, the three *Bacillus* show differences in the balance between speed selection and the random term (*Appendix 2—figure 6b*): while *B. pumilus* is slightly more influenced by the friction term than by stochasticity, these mechanisms are perfectly balanced in *B. sphaericus* accelerations, while *B. cereus* is more influenced by the random term.

## Interpretation of the bacterial swimming at the light of their morphology

Kinematics descriptors and swimming parameters can then be reinterpreted through the insights provided by the morphology of each bacteria species as shown in *Figure 2*. As observed in *Figure 2*, *B. pumilus* and *B. sphaericus* are flagellated whereas *B. cereus* is equipped by a unique brush-like bundle of thin flagella at its tail. This morphology can be linked to their swimming patterns. The flagella could be linked to the run-and-tumble behaviour of *B. pumilus* and *B. sphaericus*, as shown for other flagellated bacteria such as *E. coli*, the tumbling events of which are induced by reverse rotation of the cellular motor of its multiple flagella (*Patteson et al., 2015*). Additional functional characteristics may discriminate *B. pumilus* and *B. sphaericus*, since run-and-reverse swimming is the natural behaviour of *B. sphaericus* even in the Newtonian control buffer, whereas *B. pumilus* drastically reduces its speed in high-density biofilms (*Figure 7*, a) and starts tumbling in the host biofilm (*Figure 4*). *B. pumilus* has the highest number of flagella and is the bacteria that reaches the highest speeds specially in the Newtonian buffer and in low-density areas, indicating that this characteristic may be an advantage for swimming fast in the extracellular matrix. The kind, size and disposition of the flagella bundle may help *B. cereus* swimmers to adapt their runs to their environment by changing directions to follow lower density areas (higher impact of direction selection term of the three *Bacillus* in *Appendix 2—figure 6*) or to adapt rapidly when biofilm density varies (largest $\gamma$). *B. cereus* being the bacteria with the strongest stochastic part (highest $\epsilon$, density shifted towards $A(\epsilon)$ in *Appendix 2—figure 6*), this morphology could also help the swimmer to go through the biofilm by random navigation, which helps to maintain comparable straight trajectory with or without biofilm when the stochastic part is higher than the speed selection term (*Appendix 1—figure 1*, *Appendix 2—figure 3* and *Appendix 2—figure 6*). Finally, *B. sphaericus* bacteria are much longer than the other two species, which may explain why this species is the least motile in terms of acceleration and kinematics, both in biofilms and in the Newtonian control buffer.

## Discussion

### Modelling and analysis of swimming trajectories

When analyzing microbial swimming trajectories, two general strategies can be found in the literature. The first one aims at designing statistical tests quantifying similarities with or deviations from typical motion of interest such as diffusion (*Patteson et al., 2015*). Another strategy consists in providing a generative model of the data, analyzing it (*Chepizhko and Peruani, 2013*; *Chepizhko et al., 2013*) and comparing model outputs with real data (*Koorehdavoudi et al., 2017*; *Jabbarzadeh et al., 2014*), possibly after inference. The model that is studied in this paper belong to the second category: the model includes deterministic mechanisms describing interactions with the host biofilm, together with a random correction counterbalancing the modelling errors. The parameter inference allows to interpret the data variance relatively to speed or direction adaptations to the host biofilm versus residual effects gathered in the stochastic term. This method is comparable to ANOVA-like multi-variate analysis: the parametric phenomelogical mappings between explicative co-variables and a swimming behaviour (for example the function defining speed selection from biofilm density) are gathered in the same inference problem, enabling to decompose acceleration variability between the different swimming behaviours. This integrative method allows for multi-data integration and co-analysis. Furthermore, the fitted model allows to simulate typical swimming trajectories of a given species.

### Population-wide swimming characteristics vs true-state inference

In this study, we do not aim to recover 'true' swimmer trajectories (a.e. the blue trajectory in *Appendix 2—figure 4*), that is identifying through smoothing techniques an approximation of the specific realization of the stochastic modeling and observation errors that lead to a given 'observed'

trajectory. Rather, the goal is to identify common characteristics shared by a population of trajectories by inferring the 'population-wide' parameters (the parameters $\alpha$, $\beta$, $v_0$, $v_1$, $\gamma$, and $\epsilon$) that best explain the whole set of observed accelerations in a same population of swimmers. For this reason, we did not introduce swimmer-specific terms nor individual noise: they would have increased the model accuracy, but to the price of a blurrier characterization of the species specificities.

This choice determined our inference framework. Despite several alternative options for recovering hidden states, in particular SSM (space state models) which are common in spatial ecology (*Auger-Méthé et al., 2021*), the Bayesian method we opted for is a simpler non-linear regression problem that proved to be sufficient to recover macroscopic features of swimmer trajectories and species stratification. We discuss in Appendix 3 Various inference models the different options that were tested and present in Materials and methods Sec. Inference the method for noise model selection. Among other interesting features, the Bayesian method provides confidence intervals on the final parameter estimation, and on the resulting trajectories as in *Figure 6a*.

## Predictive capabilities of the model

The deterministic terms of the model explain only half of the variance (*Table 2*). A major part of the underlying mechanisms is not correctly described by our model which is a common feature since it is a phenomenological model which only considers interactions with the underlying biofilm at a macroscopic level, without taking into account nanoscale physical mechanisms. A more detailed description of the underlying physics could have been designed as in *Martinez et al., 2014*, but it would have made more complex the analysis of the interactions between the host biofilm and the swimmer trajectories and the extraction of species-specific patterns. However, we note that our model correctly renders observations made through macroscopic trajectory descriptors, even though the inference process has not been made based on these observables. Furthermore, several repetitions of the same models with different samples of the stochastic terms give very similar values for the trajectory descriptors (see *Appendix 2—figure 5* and section Influence of inference and stochastic terms on the trajectory descriptors), showing that these descriptors are robust to stochastic perturbations. Hence, the model (2) can be used to produce synthetic data sharing the same global characteristics than the original ones specifically taking into accounts interactions between the swimmers and the host biofilm. Furthermore, these predictions also reproduce the species stratification observed in the original data using the global descriptors.

## Biological interpretation of the fitted models

The direction selection term of the equation driven by $\beta$ has little impact in the swimmer model fitted on real data. However, the parameter $\beta$ can have a sensible impact on the kinematics as shown in the sensitivity analysis, and on the trajectories in mock biofilms (*Appendix 2—figure 1*). This could indicate that direction selection based on biofilm gradients is marginally effective in real-life swimming trajectories in a biofilm matrix. On the contrary, the speed selection term is more effective for the three *Bacillus*, showing that these micro-swimmer are able to adapt their swimming velocity to the biofilm density faced during their run. This term acts as an inertial term which enhances the stochastic term to provide direction and velocity changes.

The model has been used to decipher different adaptation strategies to the host biofilm of the three species during their swim. It confirms that *B. sphaericus* are the less motile bacteria in the biofilm, with reduced speeds and adaptation capabilities as indicated by the smallest model parameter values and a stereotypic run-and-reverse behaviour inside or outside the biofilm. *B. pumilus* on the contrary drastically changes its swimming behaviour in the biofilm compared to the Newtonian control buffer, which is reflected in the model by a high amplitude between $v_0$ and $v_1$ and a high $\gamma$ that indicates a rapid adaptation for varying biofilm densities. *B. cereus* shows the highest adaptation ability to the biofilm matrix, with the highest $\gamma$ and $\beta$ reflecting biofilm-induced speed and direction changes. Furthermore, the high stochastic effects (highest $\epsilon$) higher than the speed selection term tuned by $\gamma$ (see *Appendix 2—figure 6*) allows this swimmer to conserve straight runs in the biofilm (see Appendix 2 Sec. Friction and random term in Langevin equations.) in the same way than in the control Newtonian fluid.

This characterization methodology could be used to drive species selection for improved biofilm control. Furthermore, the model can be used to predict new trajectories and the resulting biofilm

vascularization, in a similar framework as in *Houry et al., 2012*. Coupled with a model of biocide diffusion, these simulations could be used to test numerically the efficiency of mono- or multi-species swimmer pre-treatment to improve the removal of the host biofilm.

### Flagellated bacteria in polymeric solutions

Characterization of flagellated bacteria motility in polymeric solutions is a very active research area (*Martinez et al., 2014*; *Patteson et al., 2015*; *Zöttl and Yeomans, 2019*; *Qu and Breuer, 2020*; *Qu et al., 2018*). Speed and direction variations have been measured for various polymeric fluids with different visco-elastic properties. For the model bacteria *E. coli* in polymeric solutions, enhanced viscosity decreases tumbling while increased elasticity speeds up the swimmers (*Patteson et al., 2015*; *Zöttl and Yeomans, 2019*). In our experiments on the contrary, we observed decreased speeds and strong enhancement of reverse events for the flagellated *B. sphaericus* and *B. pumilus* in the biofilm compared to the Newtonian control buffer. However, the experimental set-up shows strong differences: the complex rheology of *S. aureus* biofilms may strongly differ from polymeric fluids even if under certain condition they can be considered as visco-elastic fluids (*Gloag et al., 2020*), impacting differently the swimmer behaviours. Furthermore, the physiology of the motor cell in the Gram-positive *Bacillus* differs from the one of the Gram-negative *E. coli* (*Terahara et al., 2020*; *Szurmant and Ordal, 2004*; *Subramanian and Kearns, 2019*). Finally, the particular brush-like flagella bundle of *B. cereus* may allow this species to conserve the same swimming in Newtonian and crowded environments, by adapting its swimming speed to the local density and otherwise randomly selecting swimming directions across the host biofilm. To generalize this approach to other contexts, this study should be reproduced for other swimmers and other host biofilms, together with polymeric fluids and porous media, including biochemical interactions.

## Materials and methods
### Infiltration of host biofilms by bacilli swimmers

Infiltration of *S. aureus* biofilms by bacilli swimmers were prepared in 96-well microplates. Submerged biofilms were grown on the surface of polystyrene 96-well microtiter plates with a μ clear base (Greiner Bio-one, France) enabling high-resolution fluorescence imaging (*Bridier et al., 2010*). 200 μL of an overnight *S. aureus* RN4220 pALC2084 expressing GFP (*Malone et al., 2009*) cultured in TSB (adjusted to an OD 600 nm of 0.02) were added in each well. The microtiter plate was then incubated at 30°C for 60 min to allow the bacteria to adhere to the bottom of the wells. Wells were then rinsed with TSB to eliminate non-adherent bacteria and refilled with 200 μL of sterile TSB prior incubation at 30 celsius for 24 h. In parallel, *B. sphaericus 9* A12, *B. pumilus 3* F3 and *B. cereus 10B3* were cultivated overnight planktonically in TSB at 30 °C. Overnight cultures were diluted 10 times and labelled in red with 5 μM of SYTO 61 (Molecular probes, France). After 5 min of contact, 50 μL of labelled fluorescent swimmers suspension were added immediately on the top of the *S. aureus* biofilm. All microscopic observations were collected within the following 30 min to avoid interference of the dyes with bacterial motility. Three replicates were conducted. The same protocol has been repeated without the host biofilm (control experiments): the swimmers are added to the buffer only which is a Newtonian fluid.

### Confocal laser scanning microscopy (CLSM)

The 96 well microtiter plate containing 24 hr *S. aureus* biofilm and recently added *bacilli* swimmers were mounted on the motorized stage of a Leica SP8 AOBS inverter confocal laser scanning microscope (CLSM, LEICA Microsystems, Germany) at the MIMA2 platform (https://www6.jouy.inra.fr/mima2_eng/). Temperature was maintained at 30 celsius during all experiments. 2D+T acquisitions were performed with the following parameters: images of 147.62 × 147.62 $\mu$m were acquired at 8000 Hz using a 63×/1.2 N.A. To detect GFP, an argon laser at 488 nm set at 10% of the maximal intensity was used, and the emitted fluorescence was collected in the range 495–550 nm using hybrid detectors (HyD LEICA Microsystems, Germany). To detect the red fluorescence of SYTO61, a 633 nm helium-neon laser set at 25% and 2% of the maximal intensity was used, and fluorescence was collected in the range 650–750 nm using hybrid detectors. Images were collected during 30 s (see *Table 1* for sampling period).

Bacterial swimmers navigate within a three-dimensional biofilm matrix and confocal microscope refreshment time is not small enough to allow 3D+T images. To limit 3D trajectories, a focal plane near the well edge has been selected, where the well wall physically constrains the swimmer trajectories in one direction, which select longer trajectories in the 2D plane that can be tracked in time. Therefore, experimental data are composed of two-dimensional trajectories captured between the swimmer arrival and departure times in the focal plane, and the associated 2D+T biofilm density images that change over time due to swimmer action.

To check that the host biofilm structure is identical near the well's edge compared to other 2D slices, we took 4 replicates of *S. aureus* biofilms that were imaged in 3D using a stack of 6 horizontal images, starting from $z = 0$ near the well's edge, to $z = 6\Delta z$, at the interface between the biofilm and the bulk solution. To study the between and within biofilm density variability in the horizontal images, we subsampled them with a regular Cartesian 4 × 4 grid, resulting in a 4 × 6 x(4 × 4)=384 2D images database supplemented by metadata (stack, $z$ and $x - y$ coordinate of the subsample), before computing a clustered pairwise correlation similarity matrix and a permanova.

## Transmitted electron microscopy

Materials were directly adsorbed onto a carbon film membrane on a 300-mesh copper grid, stained with 1% uranyl acetate, dissolved in distilled water, and dried at room temperature. Grids were examined with Hitachi HT7700 electron microscope operated at 80 kV (Elexience – France), and images were acquired with a charge-coupled device camera (AMT).

## Post-processing of image data

See *Figure 1* for a sketch of the datastream from microscope raw images to model inputs and *Appendix 1—figure 1* for data visualization at each step of the post-processing pipeline.

Swimmer tracking has been applied on the red channel of the raw temporal stacks with *IMARIS* software (Oxford Instruments) using the tracking function after automated spots detection to get position time-series for each swimmer. Time-series with less than 8 time steps were filtered out.

Then, swimmer speed and acceleration time-series were computed from their position by finite-difference approximations and trajectory descriptors were extracted. The *RGB* green channel corresponding to the biofilm density temporal images were converted into grayscale and rescaled between 0 and 1 (linear scalling).

Trajectory descriptors are defined as follows. The mean acceleration and speed values, distance and displacement are computed with $\|A\|_i^s = \frac{1}{T_i^s - 2} \sum_t \|A_i^s(t)\|$, $\|V\|_i^s = \frac{1}{T_i^s - 1} \sum_t \|V_i^s(t)\|$, $dist_i^s = \Delta t \sum_{T_{0,i}^s}^{T_{end,i}^s - \Delta t} \|V_i^s(t)\|$ and $disp_i^s = \|X(T_{end,i}^s) - X(T_{0,i}^s)\|$. To compute the visited area, each trajectory piece was subsampled by computing $X_i^s(t_k) = \frac{k}{n_s} X_i^s(t) + (1 - \frac{k}{n_s}) X_i^s(t + \Delta t)$ for $k = 0, n_s$, with $n_s = 10$ and the pixels included in the ball $B(X_i^s(t_k), r)$ with radius $r = 2$ were labeled. The total area of the labelled pixels is defined as the visited area of the swimmer of species $s$.

To assess run-and-tumble behaviour, the angle $\theta_i^s(t)$ and the mean velocity $\bar{V}_i^s(t)$ between two consecutive speed vectors are defined with $\theta_i^s(t) = \arccos((V_i^s(t) \cdot V_i^s(t - \Delta t))/(\|V_i^s(t)\|\|V_i^s(t - \Delta t)\|))$ and $\bar{V}_i^s(t) = (\|V_i^s(t)\| + \|V_i^s(t - \Delta t)\|)/2$, for $t \in (T_{0,i}^s + \Delta t, T_{end,i}^s)$.

Post-processed data are available at https://forgemia.inra.fr/bioswimmers/swim-infer/SwimmerData.

## Computation of the forward swimming model

Time integration of *equations (2)* has been solved with an explicit Euler scheme regarding positions $\mathbf{x}_{i,t}^s$ and velocities $\mathbf{v}_{i,t}^s$ of the swimmer of species $s$ at time $t$:

$$\mathbf{x}_{i,t+1}^s = \mathbf{x}_{i,t}^s + \mathbf{v}_{i,t}^s dt \tag{7}$$

$$\mathbf{v}_{i,t+1}^s = \mathbf{v}_{i,t}^s + \mathbf{dv}_{i,t}^s \tag{8}$$

where $\mathbf{dv}_{i,t}^s$ is given by *Equation 2*, and depends on $\theta^s$, $V_{i,t}^s$, $x_{i,t}^s$, $b(t, x_{i,t}^s)$ and $\nabla b(t, x_{i,t}^s)$. In practice, the biofilm density and gradient maps $b$ and $\nabla b$ are discretized with a Cartesian grid corresponding to the image pixels.

During random walks, swimmer may exit the biofilm domain. When the swimmer reaches the domain boundary, a new swimmer is introduced with a velocity oriented towards the interior of the domain while the original trajectory is stopped at the boundary.

## Sensitivity analysis

A local sensitivity analysis (*Figure 1*) is performed by comparing basal simulation obtained with $\gamma = \beta = \epsilon = 1$ ($v_0$ and $v_1$ where taken as in *Appendix 1—table 3*) with 3 simulations where $\gamma$, $\beta$ and $\epsilon$ are alternatively set to 0, resulting in 3 alternative models where the speed or the direction selection or the random term is turned off. The interaction between the speed selection term (set by $\gamma$) and the random term is illustrated in *Appendix 2—figure 3* where 5 repetitions of the same trajectory of a simplified Langevin *equation (11)* are displayed with or without friction ($\gamma = 1$ or $\gamma = 0$), but with the same random seed for the stochastic term so that the stochastic part is strictly identical.

To analyze the impacts of the non-dimensionalized swimming parameters $\gamma$, $v_0$, $v_1$, $\beta$, $\epsilon$ on the loco-motion behaviour, a global sensitivity analysis has been performed. The parameter space $[0, 1]^5$ was uniformly sampled with n = 1000 points using the Fourier Amplitude Sensitivity Test (FAST) sampler of the *SALib* library that is the function *SALib.sample.fast_sampler.sample* (*Cukier et al., 1973*; *Saltelli et al., 1999*). We note that the interval $[0, 1]$ covers a large parameter domain for some parameters, in particular $\beta$ which remains small after inference. For this parameter, the sensitivity analysis will show potential impact on the output, that may be ineffective in the parameter range of the inferred model.

For each point in the parameter space, a forward simulation is conducted on a population of swimmers on a representative biofilm extracted from the dataset (first batch of the *B. pumilus* dataset). Trajectory descriptors are then extracted and taken as observable of the sensitivity anaylsis that requires both the parameters sampling and the associated descriptors. Sobol indices of first order are then returned and pairwise partial correlations matrix has been calculated. Convergence of the Sobol indices has been checked by taking sub-samples containing less than $1,000$ points.

## Inference

### Numerical implementation

The inverse problem (4)-(6) has been implemented using a Hamiltonian Monte Carlo (HMC) method to solve this Bayesian inference problem.

The three replicates for each swimmer species are pooled (trajectories and biofilm density maps) and the input data required for the inference procedure (velocity $\mathbf{yV}$ and acceleration $\mathbf{yA}$ times series for the whole batch of swimmers, biofilm densities $yb$ and gradient $\mathbf{yGb}$ extracted at swimmer positions) were assembled in a customed data structured. Normal standard prior distributions were set for all swimming parameters $\theta = (\gamma, v_0, v_1, \beta, \epsilon)$. Additional positivity constrained were imposed for all parameters but $\beta$. Therefore, the implemented model can be summarized as:

$$\theta \sim \mathcal{N}(0, 1), \quad \gamma \geq 0, \quad v_0 \geq 0, \quad v_1 \geq 0, \quad \epsilon \geq 0$$

$$\mathbf{yA} \sim \mathcal{N}(f_A \left( \gamma, v_0, v_1, \beta | yb, \mathbf{yV}, yb, \mathbf{yGb}, dt \right), \epsilon)$$

A *warmup* of 1000 runs is followed by the Markov chains construction (4,000 iterations for 4 Markov chains). Markov chain convergence is assessed by direct visualization (*Appendix 1—figure 4*) by checking for biaised covariance structures in pair-plots (*Appendix 1—figure 5*). Standard convergence index were additionnaly computed: effective sample size per iteration ($n_{eff}$) and potential scale reduction factor ($R_{hat}$).

### Noise model selection

Different noise models have been evaluated for the regression model (5) to take into account batch or individual effects. Namely, we decomposed the noise in *Equation 5* by replacing $\eta^{\mathbf{s}}$ by $\eta^{\mathbf{s}}_i$ and/or $\eta^{\mathbf{s,b}}$ for individual and experimental batch $b$. Model selection has been conducted by computing the WAIC for the different noise models. A huge degradation of the WAIC has been observed for individual or batch dependant noises, indicating that the enhancement of the inference accuracy provided by the additional parameters can be considered as over-fitting and discarded.

## Inference validation on synthetic data

### Ground truth data construction

Ground truth synthetic data (see section Assessment of the inference with synthetic data) were computed by solving *Equations 2; 8* with $\gamma = 10$, $v_0 = 5$, $v_1 = 1$, $\beta = 10$, $\epsilon = 40$ and biofilm maps taken from the first batch of the *B. pumilus* dataset. The number of swimmers was fixed to $N = 50$ and the number of time steps was taken identical to the experimental data that is $N_t = 224$. Resulting mean speeds and accelerations were $A_{ref} = 68.29$, $V_{ref} = 7.47$ and were used to rescale the data before inference together with the ground truth parameters (*Appendix 1—table 3*). In total, the acceleration dataset contains 9,523 samples for each spatial direction.

### Comparing ground truth data with the fitted model

After inference, a new dataset is obtained by solving *Equation 8* with the fitted parameters. The same initial conditions for speeds and positions as the ground truth data are taken. Trajectories are stopped after the same number of time step as in the corresponding trajectory of the ground truth dataset. To discard spurious stochastic uncertainties, the same random seed as the ground truth simulations was taken, so that the unique uncertainty source was inference errors.

### Checking the sensitivity to biofilm image noise

To produce *Appendix 1—figure 6*, the biofilm density and the biofilm density gradient maps have been noised with an additive gaussian noise with increasing variance, before inference: we set

$$\epsilon_b \sim \mathcal{N}(0, \sqrt{l}\sigma_b) \text{ and } \epsilon_{\nabla b} \sim \mathcal{N}(0, \tfrac{\sqrt{2l}}{\Delta x}\sigma_b)$$

where $\sigma_b$ is the variance observed in the original data, and $\epsilon_b$ and $\epsilon_{\nabla b}$ are respectively the noise applied to the biofilm density and the biofilm density gradient. The parameter $l \in [0, 0.01, 0.02, 0.03, 0.04, 0.05]$ is increased to apply a noise from 0% to 5%.

## Inference validation on experimental data

### Comparing microscopy data with the fitted model

The same procedure is repeated on the microscopy data: after inference, a new dataset is obtained by solving *Equation 8* with the fitted parameter, taking the same initial conditions for speeds and positions. Trajectories are stopped after the same number of time step as in the corresponding trajectory of the ground truth experimental dataset.

### Measuring the deterministic reconstruction

The deterministic coefficient of determination $R^2_{det}$ was computed to measure how much the dataset is explained by the deterministic part of the model. Setting $A_i^{s,det} = f_A\left(\gamma, v_0, v_1, \beta|yb, \mathbf{yV}, yb, \mathbf{yGb}, dt\right)$:

$$R^{2,s}_{det} = 1 - \frac{\sum_i (yA_i^s - A_i^{s,det})^2}{\sum_i (yA_i^s - y\bar{A}^s)^2}$$

where $y\bar{A}^s$ is the acceleration mean. $R^{2,s}_{det}$ is expected to tend towards 1 when the stochastic term $\eta = \mathcal{N}(0, \epsilon)$ becomes negligible with respect to $A^{det}$.

## Plots and statistics

To allow inter-species comparisons in plots, the data and model outputs are re-normalized with common reference values $A_{ref}$ and $V_{ref}$ defined as the average of the species reference values (see *Table 2* for values). Uni-dimensional distributions (*Figure 5* upper panel, *Figure 6b* upper panel, *Figure 7a*, upper panel, and *Figure 7b*) were obtained with the *gaussian_kde* function of scipy.stats. T tests for mean comparison were performed using scipy.stats *ttest_ind*.

Two-dimensional distribution plots (*Figures 5 and 6* b, *Figure 7a* lower panels) were obtained by first plotting the two-dimensional point cloud and approximating the point distribution with a gaussian KDE using scipy.stats *gaussian_kde* function. Then, the gaussian kde is evaluated at each point of the point cloud and quantiles 0.05, 0.5, and 0.95 of the resulting values are computed. Finally, quantile isovalues are plotted and the point cloud and the KDE are removed (see *Appendix 4—figure*

*1* and Sec. KDE computation for details): this procedure ensures to enclose 5, 50% and 95% of the original points, centered in the densest zones of the initial point cloud.

Ternary plots (*Appendix 2—figure 6*) were obtained by first computing the contribution of each term of *equation (4)* to acceleration estimate. Namely, note

$$s(b)_i^s = \|\gamma(v_0^s + b(t, X_i^s(t))(v_1^s - v_0^s) - \|V_i^s(t)\|)\frac{V_i^s(t)}{\|V_i^s(t)\|}\|,$$

$$s(\Delta b)_i^s = \left\|\beta^s \frac{\Delta b(t, X_i^s(t))}{\|\Delta b(t, X_i^s(t))\|}\right\|, \text{ and } s(\eta)_i^s = \|\eta^s\|$$

We compute the proportions $A(k)_i^s$ for $k \in \{b, \nabla b, \eta\}$,

$$A(k)_i^s = \frac{s(k)_i^s}{s(b)_i^s + s(\nabla b)_i^s + s(\eta)_i^s}.$$

Points $(A(b)_i^s, A(\nabla b)_i^s, A(\eta)_i^s)$ are then plotted in ternary plots using the Ternary python package (*Weinstein et al., 2019*) and approximated by gaussian KDE. Isolines are finally plotted as previously described.

To construct the plot in *Appendix 1—figure 2*, pairwise correlation of the biofilm density in the 384 samples has been computed (scikit-learn *pairwise_distances*, 'correlation' metric parameter *Pedregosa et al., 2011*), and the resulting similarity matrix has been displayed using Seaborn package *clustermap* function (*Waskom, 2021*) after hierarchical clustering (scipy.cluster.hierarchy linkage function *Virtanen et al., 2020*). Additional permanova has been computed to assess the significance of between-group dissimilarities using stats.distance package *permanova* function (*scikit-bio development team, 2020*).

## Code availability

All the image pre- and post-processing, calculations and statistics have been performed with custom scripts using the standard python libraries numpy (*Harris et al., 2020*), scipy (*Virtanen et al., 2020*), imageio (*Klein, 2021*), and pandas (*McKinney, 2010*). The forward swimming problem computation is computed using customed scripts built upon numpy (*Harris et al., 2020*) and H5py (https://www.h5py.org). Sensitivity analysis has been conducted with the *SALib* library (*Cukier et al., 1973*; *Saltelli et al., 1999*) (Sobol index, function *SALib.analyze.fast.analyze*) and the *pingouin* library (*Vallat, 2018*) (PCC, *pcorr* method). The Bayesian inference has been conducted using the *STAN* library (*Stan Development Team, 2018*) through its python interface *pystan* (*Riddell et al., 2021*). All plots have been made with the matplotlib python library (*Hunter, 2007*).

The whole *python* code have been made available and accessible at the following git repository https://forgemia.inra.fr/bioswimmers/swim-infer.

## Acknowledgements

This work has benefited from the facilities and expertise of MIMA2 MET – GABI, INRAE, AgroParistech, 78,352 Jouy-en-Josas, France. C Péchoux is warmly acknowledged for TEM observations. Financial support was provided by the French National Research Agency ANR-12-ALID-0006. Guillaume Ravel received funding from the Mathnum department at INRAE.

## Additional information

### Funding

| Funder | Grant reference number | Author |
|---|---|---|
| Mathnum department - INRAe | | Guillaume Ravel |
| Agence Nationale de la Recherche | ANR-12-ALID-0006 | Romain Briandet |

| Funder | Grant reference number | Author |
|--------|------------------------|--------|

The funders had no role in study design, data collection and interpretation, or the decision to submit the work for publication.

## Author contributions

Guillaume Ravel, Formal analysis, Investigation, Methodology, Project administration, Software, Visualization, Writing – original draft, Writing – review and editing; Michel Bergmann, Methodology, Validation, Writing – review and editing; Alain Trubuil, Conceptualization, Methodology, Validation, Writing – review and editing; Julien Deschamps, Data curation, Validation, Writing – review and editing; Romain Briandet, Conceptualization, Data curation, Funding acquisition, Investigation, Methodology, Validation, Writing – review and editing; Simon Labarthe, Conceptualization, Formal analysis, Funding acquisition, Investigation, Methodology, Project administration, Software, Supervision, Validation, Writing – original draft, Writing – review and editing

## Author ORCIDs

Romain Briandet (iD) http://orcid.org/0000-0002-8123-3492
Simon Labarthe (iD) http://orcid.org/0000-0002-5463-7256

## Decision letter and Author response

Decision letter https://doi.org/10.7554/eLife.76513.sa1
Author response https://doi.org/10.7554/eLife.76513.sa2

# Additional files

## Supplementary files

• Transparent reporting form

## Data availability

Data and code have been deposited at https://forgemia.inra.fr/bioswimmers/swim-infer and https://doi.org/10.5281/zenodo.6560673.

The following dataset was generated:

| Author(s) | Year | Dataset title | Dataset URL | Database and Identifier |
|-----------|------|---------------|-------------|-------------------------|
| Labarthe S, Ravel G, Deschamps J, Briandet R | 2022 | Inferring characteristics of bacterial swimming in biofilm matrix from time-lapse confocal laser scanning microscopy: compagnon code and data | https://doi.org/10.5281/zenodo.6560673 | Zenodo, 10.5281/zenodo.6560673 |

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

## Appendix 1

### Illustration of the datastream
#### Data acquisition

Illustrations of the image data at different steps of the data stream are displayed in **Appendix 1—figure 1**, from raw microscopy data to rescalled biofilm density map with trajectories. The contrast of the original 2 chanel image has been enhanced for visualization. The *RGB* biofilm density temporal images (see Materials and methods) were converted into grayscale and rescaled between 0 and 1 (linear scalling). In this images, for illustrations, trajectories are mapped into the biofilm density map and rescaled density map at initial condition of the first *B. pumilus* batch. In the dataset, the trajectories are associated with the corresponding biofilm map: $X_i^s(t)$ is associated with the value $b(t, X_i^s(t))$ for swimmer  of species $s$ at time $t$. As the biofilm density map is also a time-series, the trajectories can hardly be represented on the underlying biofilm that also changes in time.

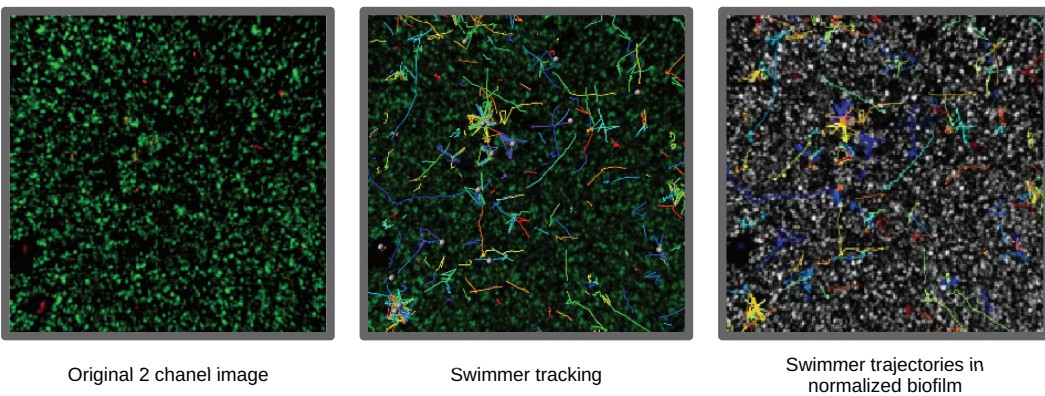

| Original 2 chanel image | Swimmer tracking | Swimmer trajectories in normalized biofilm |

**Appendix 1—figure 1.** Illustration of image data along the post-processing process. Raw data (2 chanel images) are first displayed. Then, trajectory tracking are obtained. Finally, the biofilm density map is rescalled, and mapped to grayscale. Images dimensions are 147x147μm.

### Assessing the 3D structure of the biofilm

We check that the selection of a 2D focal plan does not induce an additional bias by over-selecting biofilm areas with specific structures near the well's edge. To do so, we assembled an additional dataset of 4 replicates of *S. aureus* 3D images (see Materials and methods, section 4.2, and **Appendix 1—figure 2**.A for the dataset assembly) of horizontal image subsamples, and computed their within and between dissimilarities (see Materials and methods), section Plots and statistics. The resulting pairwise correlation matrix is displayed in **Appendix 1—figure 2** after hierarchical clustering. It shows that the $z$ direction does not structure the information, since the images are not clustered according to their $z$ coordinates contrary to the stack or the $x - y$ coordinate labels. Permanova analysis shows that the differences between stacks and $x - y$ subsamples are significant ($p - value = 1e - 4$) but not between horizontal images ($p - value = 1$).

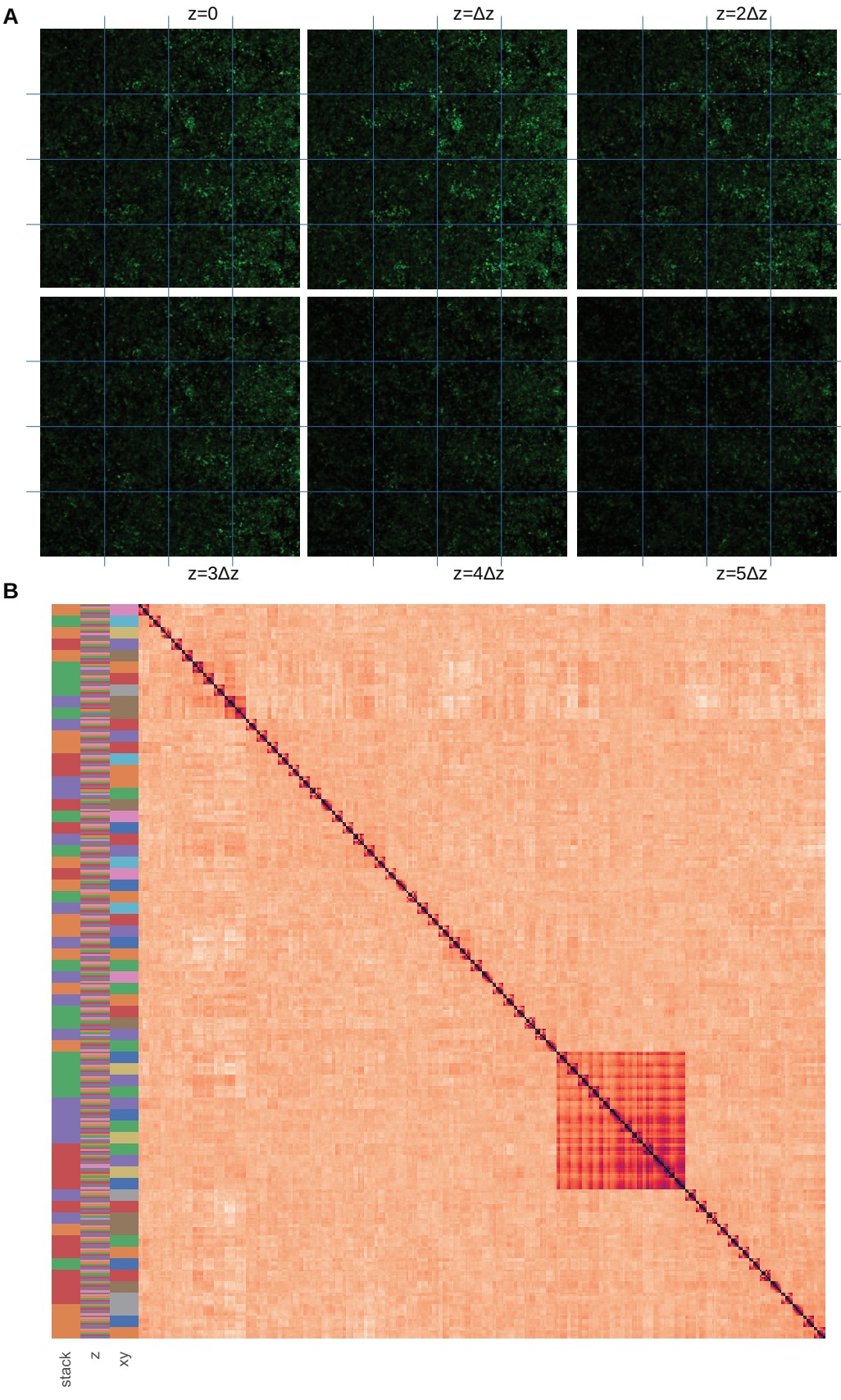

*Appendix 1—figure 2 continued on next page*

*Appendix 1—figure 2 continued*
**Appendix 1—figure 2.** Assessment that the biofilm structure does not strongly vary in the $z$ direction.
(**A**) Subsampling procedure. We illustrate the subsampling procedure in one of the 4 replicates. The 2D images constituting the 3D stack are sampled with a 4 × 4 cartesian grid. We can also visually observe that the biofilm variation between horizontal images are weak. Images dimensions are 147x147µm. (**B**) Pairwise correlation matrix. The correlation dissimilarity between sample pairs is displayed (black = 0 value, indicating identical samples, to light orange >1, indicating dissimilar samples) after hierarchical clustering. We indicate the stack, $z$ and $x - y$ label in the 3 first columns with a color code. We can observe that the samples are not gathered by, $z$ but rather by stacks and $x - y$ groups, indicating that images with identical $x - y$ labels are clustered together, showing that they are more similar to samples with the same $x - y$ coordinates in other $z$ slices, than other samples in the same $z$ slice with other $x - y$ coordinates.

## Illustration of pore formation

As strongly documented in *Houry et al., 2012*, swimmers can dig pores in a exogenous biofilm, which enhance the biofilm innervation and facilitate the penetration of macromolecules. To illustrate the pore formation, we show two successive images taken from a 2D temporal stack of *B. sphaericus* swimmers in a *S. aureus* host biofilm in *Figure 3*. In the dashed ellipse, we can see a swimmer that has moved in the two successive images, letting behind it an empty space free from host bacteria.

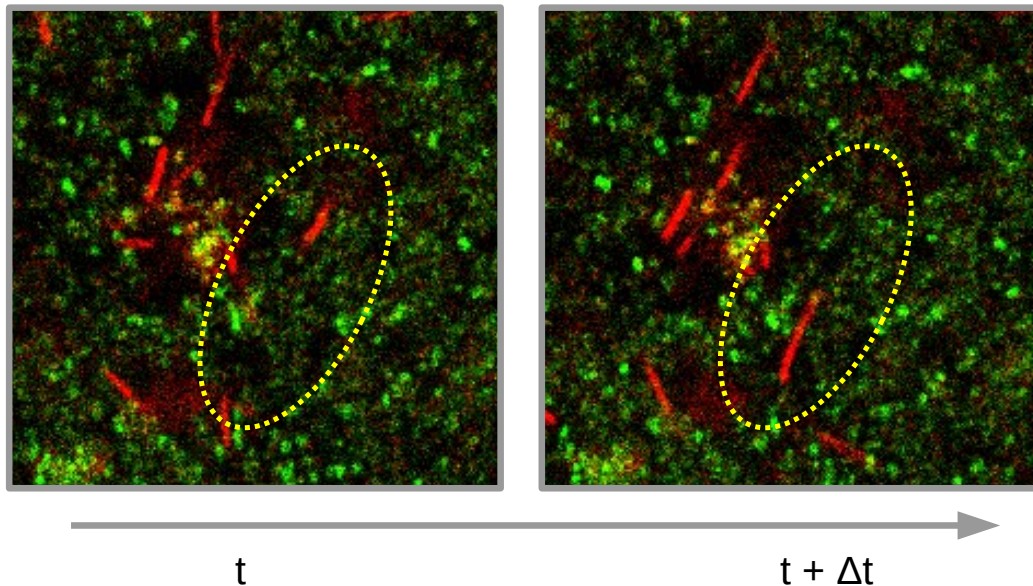

t                    t + Δt

**Appendix 1—figure 3.** Illustration of pore formation. Extractions of two successive images of *B. sphaericus* swimming in a *S. aureus* biofilm are displayed. The dashed ellipse indicates a zone where a swimmer moves between the two successive images, which creates a pore along its swimming path. Images dimensions are 76 x 78 µm.

## Statistical tests

T-tests were performed to compare mean differences between 1D distribution of *Figure 5*. Resulting p-values are displayed in *Appendix 1—table 2*.

**Appendix 1—table 1.** P-values of pairwise comparison between distributions in biofilms. Pairwise comparison were performed between 1D distributions displayed in *Figure 5* using T-test and p-values are displayed. Non-significant values are indicated in bold.

| | ||A|| | | ||V|| | | dist | | disp | | Area |
|---|---|---|---|---|---|---|---|---|---|---|
| | B. cereus | B. sphaericus | B. cereus | B. sphaericus | B. cereus | B. sphaericus | B. cereus | B. sphaericus | B. cereus | B. sphaericus |
| B.pumilus | $5.e-9$ | $1.e-10$ | $4.e-2$ | $1.e-13$ | $1.e-5$ | $2.e-3$ | $8.e-3$ | $7.e-10$ | $7.e-1$ | $4.e-14$ |
| B.cereus | | $5.e-51$ | | $2.e-13$ | | $3.e-1$ | | $5.e-15$ | | $1.e-12$ |

**Appendix 1—table 2.** P-values of pairwise comparison between distributions in the control Newtonian buffer.
Pairwise comparison were performed between 1D distributions displayed in *Figure 5* using T-test and p-values are displayed. Non-significant values are indicated in bold.

| | ||A|| | | ||V|| | | dist | | disp | | Area |
|---|---|---|---|---|---|---|---|---|---|---|
| | B. cereus | B. sphaericus | B. cereus | B. sphaericus | B. cereus | B. sphaericus | B. cereus | B. sphaericus | B. cereus | B. sphaericus |
| B.pumilus | $3.e-11$ | $9.e-4$ | $9.e-3$ | $2.e-2$ | $1.e-10$ | $2.e-2$ | $2.e-21$ | $1.e-14$ | $4.e-2$ | $6.e-1$ |
| B.cereus | | $9.e-1$ | | $2.e-7$ | | $2.e-12$ | | $8.e-1$ | | $1.e-1$ |

## Assessment of the inference with synthetic data

**Appendix 1—table 3.** Inference results on synthetic data.
The normalized ground-truth parameter values (*i.e.* ground truth parameter rescaled with $A_{ref}$ and $V_{ref}$) are compared with the inference outputs on synthetic data: posterior distribution mean and standard deviation are indicated, together with the inferred confidence intervals for the true parameters. Convergence diagnosis indices are also given, with $n_{eff}$ the effective sample size per iteration and $R_{hat}$ the potential scale reduction factors, indicating that convergence occurred for all parameters.

| parameter | ground truth | mean | std | confidence interval [2.5%–97.5%] | $n_{eff}$ | $R_{hat}$ |
|---|---|---|---|---|---|---|
| $\gamma$ | 1.094 | 1.08 | $1 \times 10^{-2}$ | $[1.06 - 1.1]$ | 3,569 | 1.0 |
| $v_0$ | 0.669 | 0.66 | $1 \times 10^{-2}$ | $[0.64 - 0.68]$ | 3,710 | 1.0 |
| $v_1$ | 0.134 | 0.13 | $2 \times 10^{-2}$ | $[0.09 - 0.17]$ | 3,431 | 1.0 |
| $\beta$ | 0.146 | 0.16 | $6, 2 \times 10^{-3}$ | $[0.15 - 0.17]$ | 5,050 | 1.0 |
| $\epsilon$ | 0.586 | 0.59 | $3 \times 10^{-3}$ | $[0.58 - 0.59]$ | 4,906 | 1.0 |

To assess the inference method, synthetic data are built and will be used as reference for assessment. We arbitrarily fix a parameter vector and solve system (1) from random initial positions, in a host biofilm arbitrarily chosen in the image dataset. We then extract the swimmer positions at given time-steps and recover accelerations and speeds with the same post-processing pipeline as for microscopy images and solve the inverse problem (5)-(6). If the inference process correctly works, we expect to recover the original parameters (the ground truth).

The ground truth parameters are correctly recovered by the inference procedure (*Appendix 1—table 3*), indicating that the parameters are correctly identifiable and that the inverse problem is well-posed. An error of respectively 1.28, 1.34, 2.98% and 0.68% on the parameters $\gamma$, $v_0$, $v_1$ and $\epsilon$ is observed in this controlled situation, $\beta$ being inferred with lower accuracy (9.59 %). This estimate is robust to noise on the biofilm data, with highest impact on $\beta$ (*Appendix 1—figure 6*). To assess the impact of parameter inference uncertainties on trajectory computation, the posterior parameter distribution is sampled and new trajectories are computed, replacing the ground-truth parameters by the sampled ones. The swimmer ground truth trajectories are accurately recovered: the sampled trajectories tightly frame the original swimmer path as illustrated on a randomly chosen trajectory (*Figure 6a*). We note that an identical random seed has been taken for these simulations, including the ground truth trajectory, in order to turn off the stochastic uncertainties and only focus on the propagation of inference errors during simulations of swimmer trajectories.

Finally, we re-assemble a synthetic dataset by replacing the ground-truth parameters by the inferred ones, that is the posterior mean. Qqplot of the fitted model accelerations versus the ground truth accelerations give an excellent accuracy (*Figure 6d–e*), with all the points lying on the bisector, except slight divergences on the distribution tails. The fitted model trajectories visually reproduce the qualitative characteristics of the original dataset (*Figure 6c*). The trajectory descriptors of section Characterizing bacterial swimming in a biofilm matrix through image descriptors are then computed on both datasets (ground truth and inferred) and compared (*Figure 6b*). The kinematics descriptors, that is acceleration and speed distributions, are very accurately recovered with a relative error of 0.1%, 3.2%, 5% for respectively the mean, quantiles 0.05 and 0.95 of the acceleration (resp. 0.9%, 2.5%,2% for speed). Some small discrepancies can be observed on the distance and displacement distributions, even if the mean and the quantiles 0.05 and 0.95 are close. The interactions between the host biofilm and the acceleration and speed distribution are also recovered with high accuracy. We note that part of the observed discrepancies comes from an additional source of variability of the simulation framework: when a swimmer reaches a domain boundary during a simulation, its trajectory is stopped and a new swimmer is randomly introduced elsewhere in the biofilm (see Materials and methods for more details). This simulation strategy seems to be responsible of the over-representation of short trajectories in the inferred dataset, compared to the ground truth (*Figure 6b* upper panel, distance and displacement distributions).

## Markov chains convergence and correlation
Markov chain (*Appendix 1—figure 4*) and markov chain pairplots (*Appendix 1—figure 5*) are displayed. Direct visualization of the posterior sampling allows to detect convergence failure (strong autocorrelation or stationnary markov chain). Markov chain pairplot informs on potential correlation between different parameters posterior samples, showing an interaction between parameter and an identification issue. In *Appendix 1—figure 4*, the markov chains correctly converged for all the parameter. No strong correlation can be observed in *Appendix 1—figure 5*.

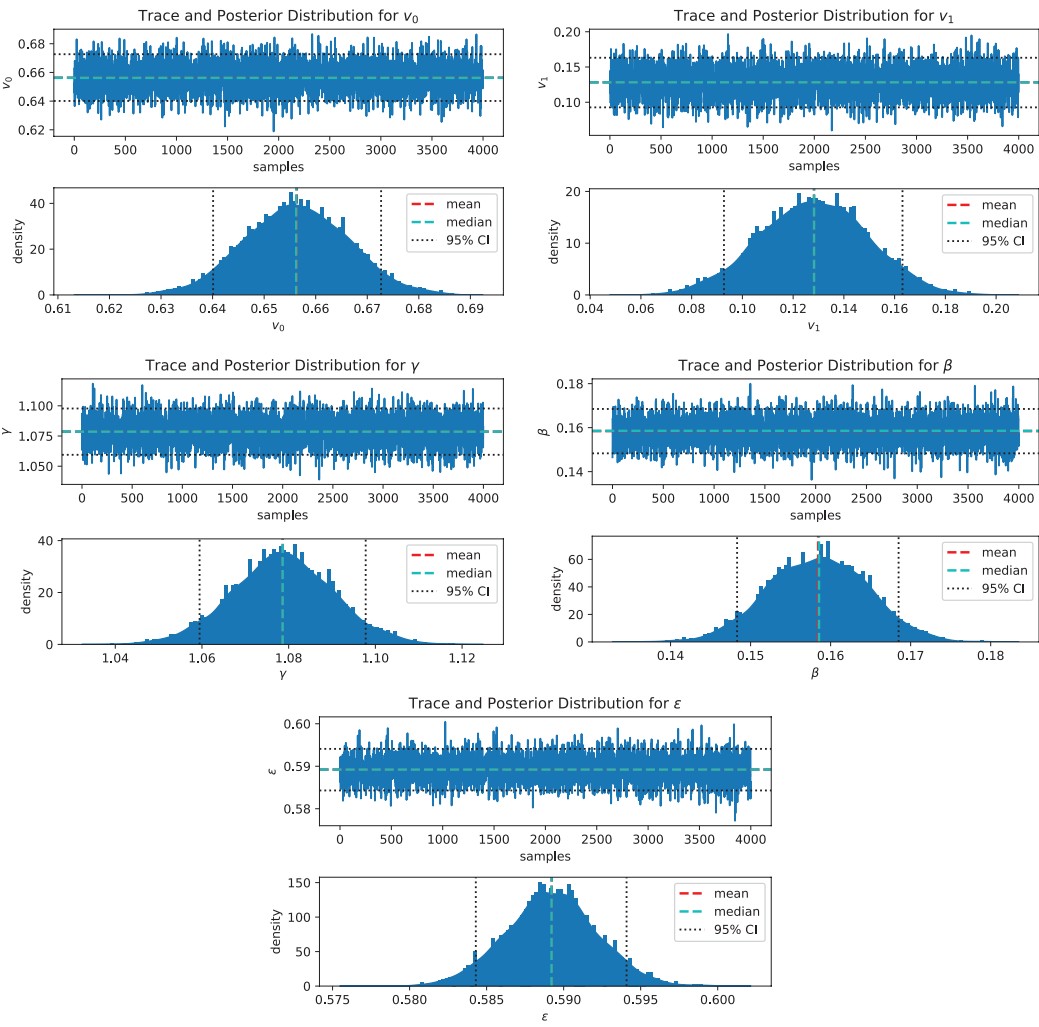

**Appendix 1—figure 4.** Inference convergence validation. The markov chain (upper panel) and the posterior distribution (lower panel) of each parameter is displayed, showing good convergence of the stochastic sampling of the posteriors.

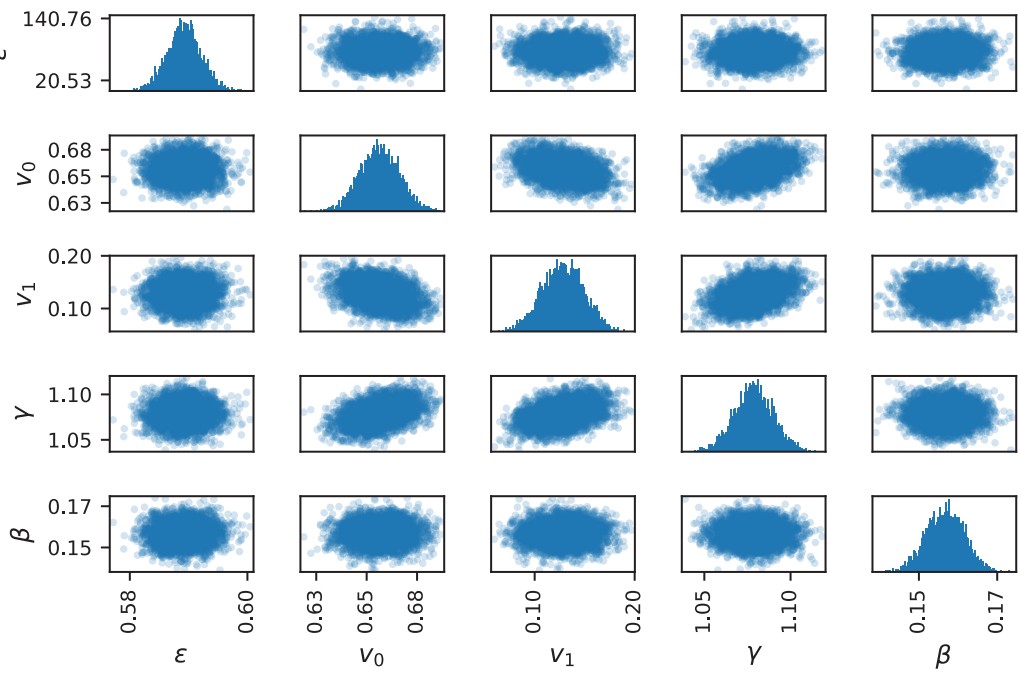

**Appendix 1—figure 5.** Pairplot of parameter markov chains. No strong covariance effect can be observed, showing that the model can not be reduced by analytical dependence between parameters. Slight correlation is observed between the parameters $v_0$, $v_1$ and $\gamma$: this feature is not surprising since $\gamma$, $v_0$ and $v_1$ are in the same term of **equation (2)**. The correlation is however too low to expect a model reduction.

## Impact of noise on biofilm data

The impact of noise on the parameter inference is assessed by noising the biofilm density and the biofilm density gradients with an additive gaussian noise with increasing variance (**Appendix 1—figure 6**). The noise variance is scaled with the variance observed in the original data. Namely, we set

$$\epsilon_b \sim \mathcal{N}(0, \sqrt{l}\sigma_b) \tag{9}$$

and

$$\epsilon_{\nabla b} \sim \mathcal{N}(0, \tfrac{\sqrt{2l}}{\Delta x}\sigma_b) \tag{10}$$

where $\sigma_b$ is the variance observed in the original data, $\epsilon_b$ and $\epsilon_{\nabla b}$ are respectively the noise applied to the biofilm density and the biofilm density gradient and $\Delta x$ is a pixel width. The parameter $l \in [0, 0.01, 0.02, 0.03, 0.04, 0.05]$ is increased to apply a noise from 0% to 5%.

We can observe that the estimate of only two parameters is impacted by noising the biofilm inputs: the estimate of $\beta$ and $v_1$. The parameter $\beta$ is also the parameter which is the less accurately inferred when no noise is added (5%). Its estimation relative error increases up to 35% when 5% noise is added. The parameter $\beta$ tunes the direction selection, which is the less effective process in the swimmer model. The other parameters are recovered with correct accuracy (kept under 18% for $v_1$, and under 6% otherwise).

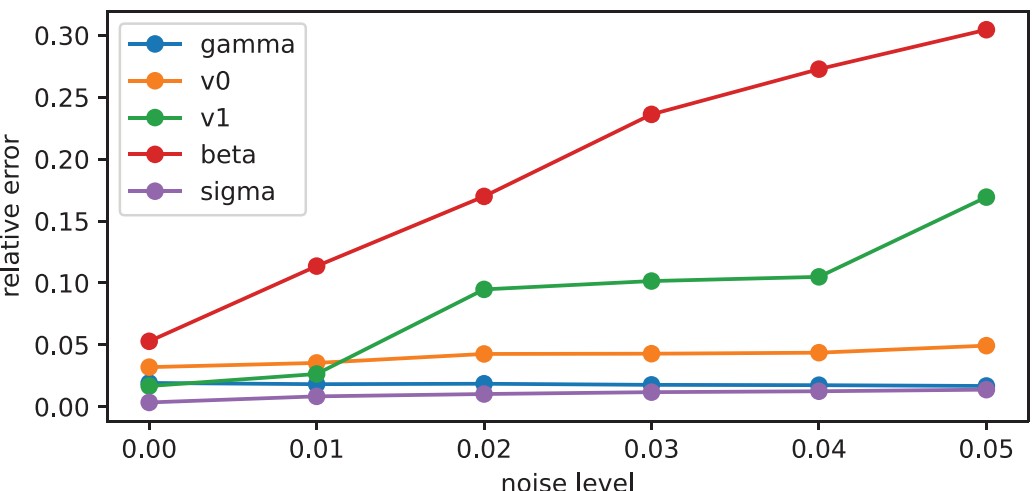

**Appendix 1—figure 6.** Impact of noise level on parameter inference. We plot the relative error of the estimate of the different equation parameters for increasing noise applied on the biofilm density and the biofilm density gradients input data.

## Appendix 2

### Numerical exploration

To illustrate the impact of each parameter on the interplay between the host biofilm and the swimmers trajectories, the model (2) was first computed on two mock biofilms. The first one is a square linear density gradient and the second is composed of large pores on a textured background mimicking the dense biofilm zones (*Appendix 2—figure 1a*). A basal simulation is computed with $\gamma = \beta = \epsilon = 1$ and will be used later on as reference for comparisons. These three parameters are alternatively set to zero to assess the resulting trajectories when the speed selection, the direction selection or the random term is shut down. Suppressing speed selection results in rectilinear trajectories ($\gamma = 0$, *Appendix 2—figure 1c*), which is rather counter-intuitive since the remaining terms are designed to tune the direction. A discussion of this phenomenon is provided in Appendix 2 Influence of inference and stochastic terms on the trajectory descriptors. When suppressing direction selection ($\beta = 0$, *Appendix 2—figure 1d*), the trajectories are no longer drifted downwards the gradient in the upper panel as in the basal simulation, and no longer follow the pores (lower panel). If the stochastic term is shut down ($\epsilon = 0$, *Appendix 2—figure 1e*), the trajectories directly go down the gradients and are trapped in the center of the image in the upper panel. When a pore is found along the run, the swimmer keeps following it without being able to escape the pore any longer unlike the basal situation (lower panel).

The link between the model parameters and the global trajectory descriptors introduced in Section Characterizing bacterial swimming in a biofilm matrix through image descriptors is less intuitive. A global sensitivity analysis of the trajectory descriptors (mean acceleration and speed, distance, displacement and visited areas) with respect to the parameters $\gamma$, $v_0$, $v_1$, $\beta$ and $\epsilon$ is conducted in Model sensitivity analysis by computing their first order Sobol index (SI) and their pairwise correlation coefficient (PCC). The sensitivity analysis shows that the mean speed is mainly influenced by $\gamma$ and $\epsilon$ with slightly negative and positive impact respectively, while acceleration is rather influenced by $\beta$ and $\epsilon$ with positive impact. The link between the parameters and the other descriptors is more complex, including non linear effects (strong SI and small PCC) and parameter interactions (higher SI residuals, see *Appendix 2—figure 2*).

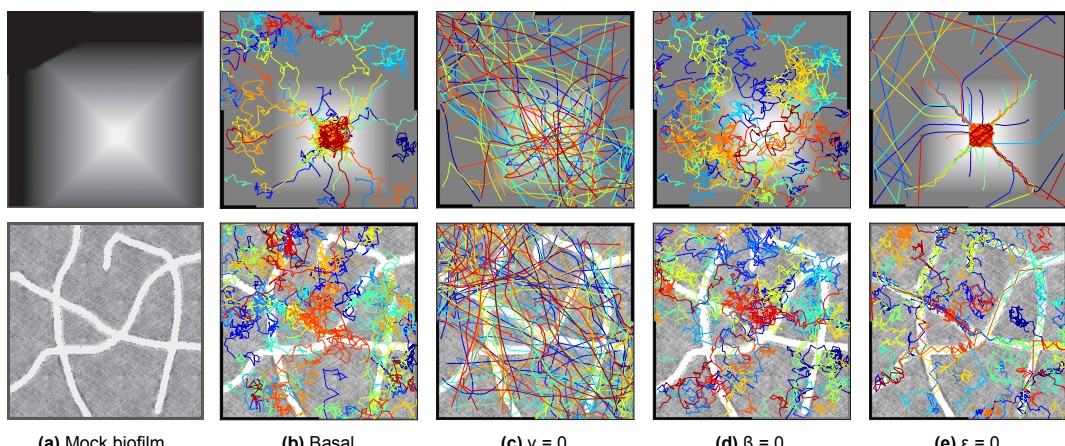

(a) Mock biofilm (b) Basal (c) γ = 0 (d) β = 0 (e) ε = 0

**Appendix 2—figure 1.** To illustrate the influence of each term of *Equation 2*, they are alternatively turned off (**c** to **e**), and swimmer trajectories are computed on mock biofilms. (**a**) displaying marked density gradients (upper pannel) or marked pores (lower pannel). Trajectories can be compared to a basal simulation (**b**) when all the terms have the same intensity (1).

### Model sensitivity analysis

The link between the model parameters and the global trajectory descriptors introduced in Section Characterizing bacterial swimming in a biofilm matrix through image descriptors is not intuitive. A global sensitivity analysis of the trajectory descriptors (mean acceleration and speed, distance, displacement and visited areas) with respect to the parameters $\gamma$, $v_0$, $v_1$, $\beta$ and $\epsilon$ is conducted by computing their first order Sobol index (SI) and their pairwise correlation coefficient (PCC).

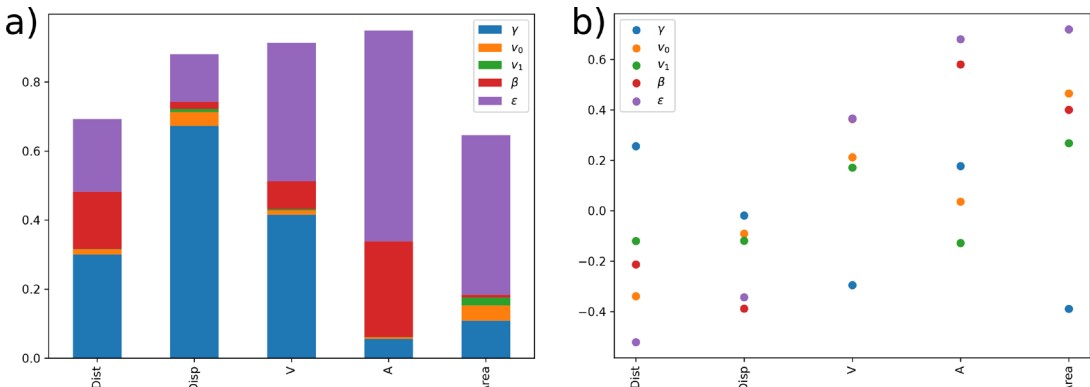

**Appendix 2—figure 2.** Sensitivity analysis of state observable respectively to state-equation parameters. The sensitivity of different state observables to parameter shifts is systematically studied through sensitivity analysis methods. (**a**) Sobol indices are displayed for each output through barplots indicating the part of variance explained by a given parameter. The bars do not reach the value 1, indicating a residual variance reflecting interactions between parameters. (**b**) Pairwise correlation coefficient (PCC) of the observable respectively to the input parameters are displayed. A negative PCC indicates a negative impact on the output, and conversely. We note that the red dot indicating the PCC of $\beta$ for $V$ is confounded with the purple one indicating the PCC of $\epsilon$.

The residual variance is small for the median speed and acceleration but slightly larger for the distance, displacement and visited area indicating larger effects of parameter interactions for these outputs, that is output variations induced by joint shifts of the parameters (*Appendix 2—figure 2*). The SI of the parameters $v_0$ and $v_1$ are negligible, except for the displacement and the visited area. The parameters $\gamma$, $\beta$ and $\epsilon$, that is the three weights associated to each component of the state *Equation 2*, are more influential. Distance and speed have several main drivers. The distance is impacted nearly equally by $\gamma$, $\beta$ and $\epsilon$ and the PCC of these parameters is quite small, indicating that these parameters may induce indistinctly negative or positive variations of the travelled distance, except for $\epsilon$ which is slightly negatively correlated. The median speed is mainly impacted by $\epsilon$ (slightly positively) and $\gamma$ (slightly negatively), with relatively small PCC (*Appendix 2—figure 2*). The mean acceleration, the displacement and the visited area are preponderantly impacted by a main driver: the mean acceleration and the visited area are particularly impacted by $\epsilon$, the stochastic term weight, with positive influence. The displacement is mainly influenced by $\gamma$ with no preponderant variation direction (null PCC, *Appendix 2—figure 2*).

## Friction and random term in Langevin equations
To illustrate the interplay between the friction and the random term during a random walks, we solve the problem

$$\mathbf{dv} = -\underbrace{\gamma\mathbf{v}\mathrm{dt}}_{friction} + \underbrace{\eta\mathrm{dt}}_{random\ term} \tag{11}$$

$$\mathbf{v}(0) = (0, 0) \tag{12}$$

$$\mathbf{X}(0) = (0, 0) \tag{13}$$

in an unconstrained domain, with $\eta$ a 2 dimensional white noise with unitary variance. The friction parameter $\gamma$ is alternatively set to 1 (*Appendix 2—figure 3*, upper panel) or 0 (*Appendix 2—figure 3*, lower panel). We note that the random seed is the same for the simulations with or without the friction term, so that the stochastic contribution is completely identical in the upper and lower panels. The trajectories produced without the friction term are much more regular and rectilinear that those produced with the friction term, that are much chaotic.

The reason of that behaviour may come from the null mean of the white noise. Roughly speaking, in average, the acceleration shows small variations around zero which leads after temporal integration to regular speeds and rectilinear-like trajectories. By contrast, the friction term reduces the particle inertia, enhancing the impact of the stochastic term, which produces much more chaotic trajectories.

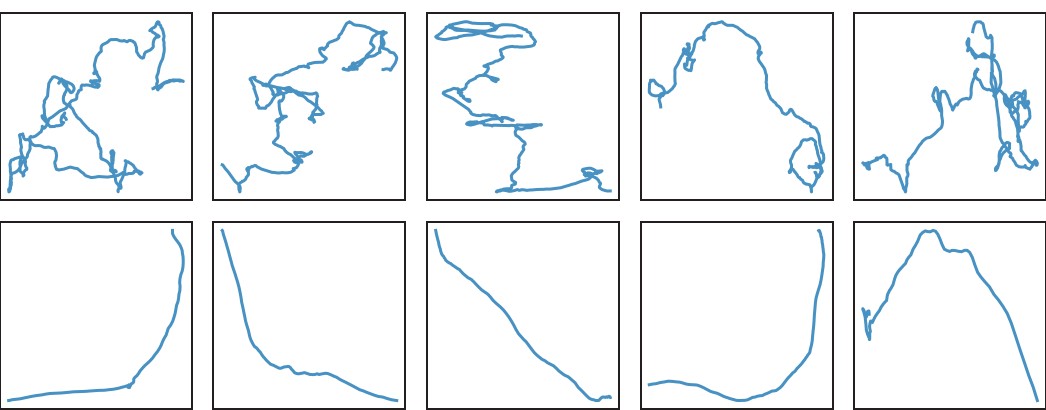

**Appendix 2—figure 3.** Illustration of the interplay between friction and stochastic terms in Langevin equation. Trajectories produced by different repetitions of *Equation 11* are displayed with $\gamma = 1$ (upper panel) and $\gamma = 0$ (lower pannel). We note that the same random seed has been taken for the simulations of the same column with or without the friction term, meaning that the stochastic term is strictly identical in both simulations.

## Impact of the stochastic term

We illustrate the impact of the random walk term on the overall swimmer trajectory with *Appendix 2—figure 4*. In this figure, we display two trajectories computed from model (2) with identical parameters ($\alpha$, $\beta$, $v_0$, $v_1$, $\gamma$ and $\epsilon$), initial condition, host biofilm and time length. Different random samplings of the stochastic term of *Equation (2)* lead to these very different trajectories. This example illustrate the difference between identifying population-wide characteristics and inferring true trajectories: while the later try to detect the differences between the two trajectories (*i.e.* in this example, identifying and smoothing the different stochastic samples leading to these trajectories), the former focuses on the common features between these apparently different trajectories.

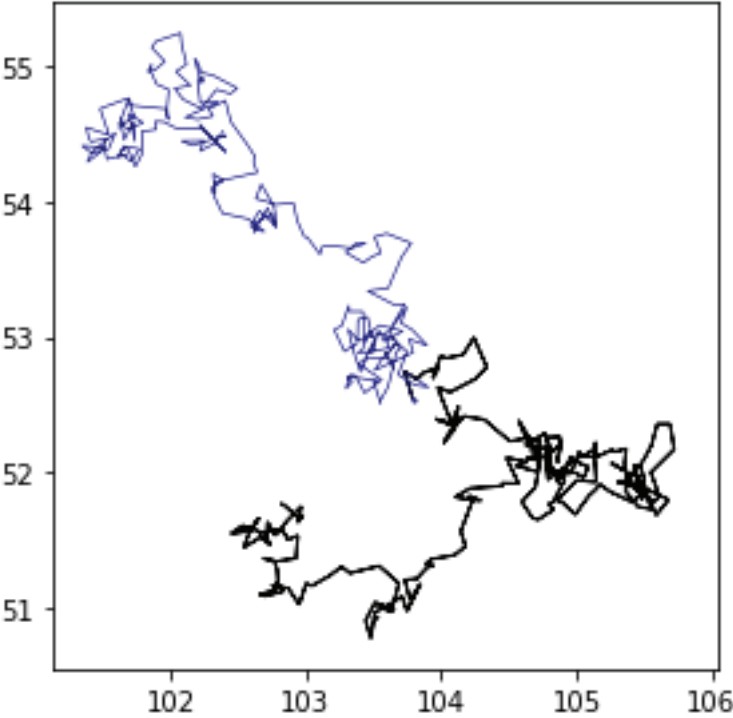

**Appendix 2—figure 4.** Influence of the stochastic process on swimmer trajectories. We plot two different trajectories computed with the model (2), including the same parameters $\alpha$, $\beta$, $v_0$, $v_1$, $\gamma$ and $\epsilon$, the same initial condition and identical host biofilm. The only uncertainty source comes from the different random samplings of the stochastic term. In this simulation, the ground truth (with default random seed) is plotted in blue.

## Influence of inference and stochastic terms on the trajectory descriptors

We wonder if the uncertainty sources involved in the inference process and in the stochastic term of the random walk have a decisive impact on the trajectory descriptors. To address this question, a first dataset is assembled by integrating in time *Equation 2* for given parameters (see *Appendix 1—table 3*), initial conditions and host biofilm. Then, this dataset is used as inputs of the inference method to infer the initial parameters (ground truth). Another dataset is produced by replacing the initial parameters by the inferred parameters. We note that we take the same seed for the random number generator than for the initial dataset, so that the only uncertainty that has been introduced until this step comes from the inference procedure. Finally, we produce a last dataset by solving the model with the same inferred parameters as in the second dataset, but changing the seed of the random number generator. Hence, this last dataset involves uncertainties coming from the stochastic terms and from the inference process. This variation results in modifying the sampling of the stochastic terms and leads to strong modifications of the trajectories, like in *Appendix 2—figure 4*.

At end, the trajectory descriptors are computed and plotted in *Appendix 2—figure 5*. We can see that the trajectory descriptor distributions are very similar across the different dataset, except for the total distance and the displacement where discrepancies can be noted. However, these differences are relatively small compared to the mean and the width of the distributions. We can also observe that the interactions with the underlying biofilm is very well conserved, even when the sampling of the stochastic term is very different. This observation grounds the initial guess that these trajectory descriptors captures common global features of the different trajectories rather than specificities of given trajectories.

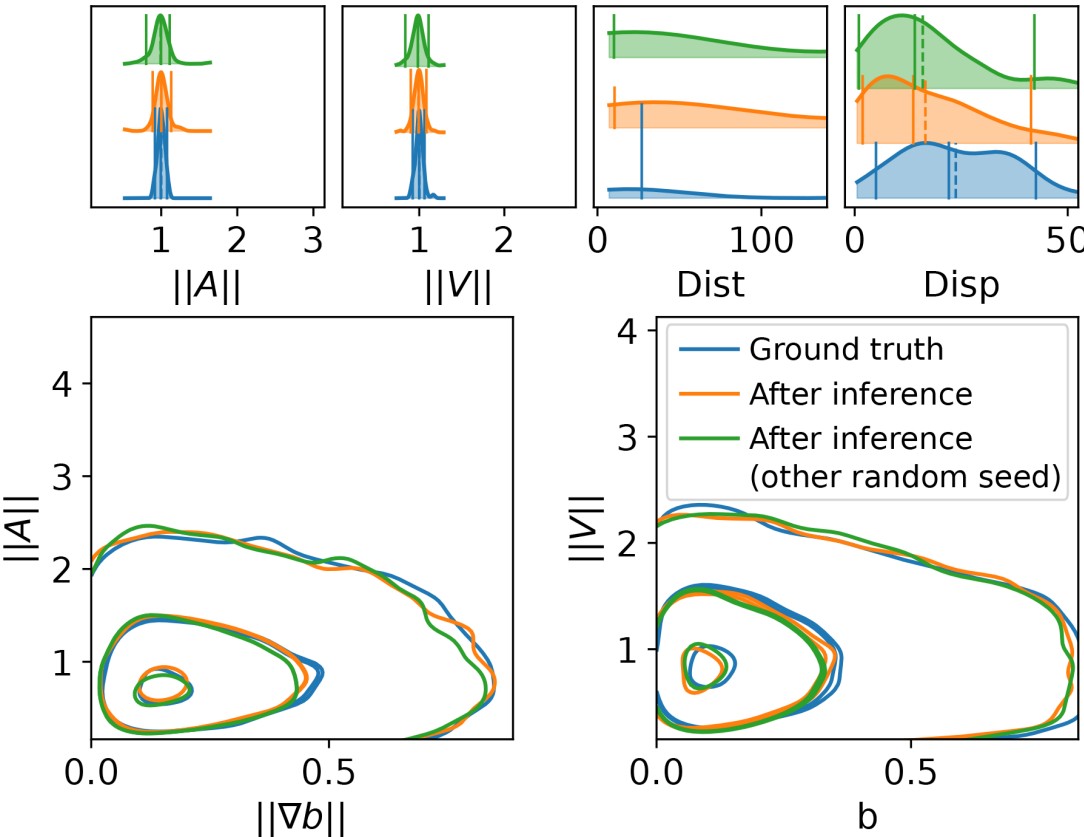

**Appendix 2—figure 5.** Low influence of the stochastic term on the trajectory descriptors. To assess the influence of the random term on the population-wide trajectory descriptors and overall prediction accuracy, we repeated the experiment displayed in *Figure 6a*. A synthetic database was first assembled (ground truth) and prediction were performed with a fitted model (After inference). Then, a second repetition of the prediction of the fitted model was computed with another seed for the random number generator, resulting in modifying the sampling of the stochastic terms and strong modifications of individual trajectories, like in *Figure 4*. The population-wide trajectory descriptors are however slightly impacted by this random effect, indicating that the main characteristics of the trajectory populations marginally depend on the stochastic term.

## Relative impact of the different swimming processes on the species swim

The ternary plot presented in *Appendix 2—figure 6* shows the balance between the different swimming processes. The contribution of each term of *equation (4)* to acceleration estimate was first computed. Namely, the relative value of the speed selection term $\|s(b)_i^s\|$, the direction selection term $\|s(\nabla b)_i^s\|$ and the stochastic term $\|s(\eta)_i^s\|$ where

$$s(b)_i^s = \|\gamma(v_0^s + b(t, X_i^s(t))(v_1^s - v_0^s) - \|V_i^s(t)\|)\frac{V_i^s(t)}{\|V_i^s(t)\|}\|,$$

$$s(\nabla b)_i^s = \|\beta^s \frac{\nabla b(t, X_i^s(t))}{\|\nabla b(t, X_i^s(t))\|}\|, \quad \text{and} \quad s(\eta)_i^s = \|\eta^s\|.$$

The proportions $A(k)_i^s$ of each process was computed with

$$A(k)_i^s = \frac{s(k)_i^s}{s(b)_i^s + s(\nabla b)_i^s + s(\eta)_i^s}.$$

for $k \in \{b, \nabla b, \eta\}$. As the contribution of the direction selection was limited compared to the other processes, we zoomed in the plot near the edge $\|s(\nabla b)_i^s\| = 0$ to allow inter-species comparisons.

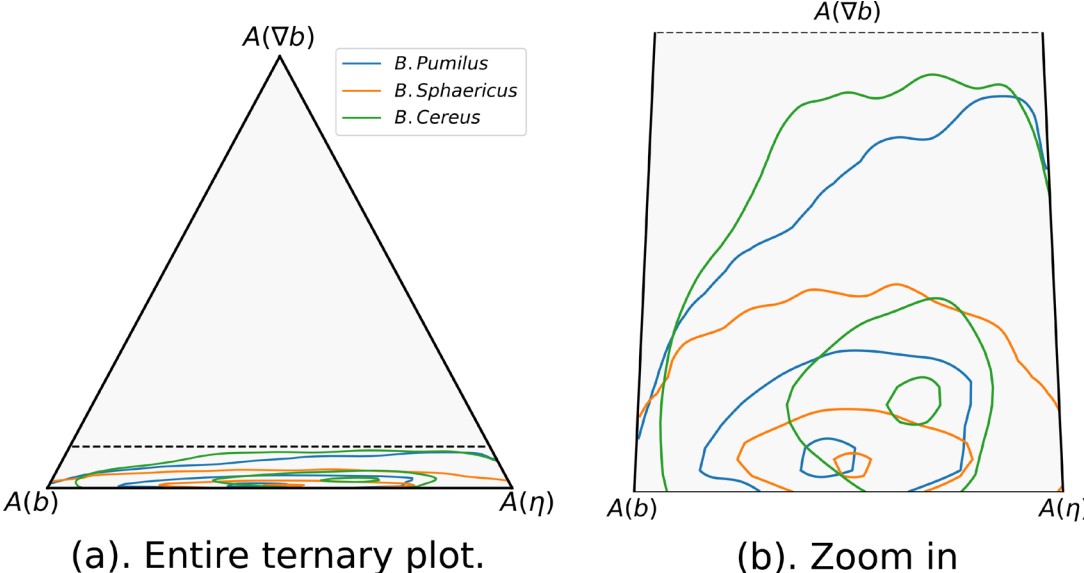

(a). Entire ternary plot. (b). Zoom in

**Appendix 2—figure 6.** Respective influence of stochastic effects, speed or direction adaptation to the host biofilm. We plot in a ternary plot the respective influence of the speed selection ($V$), the direction selection ($D$) and the random term ($\epsilon$) of **Equation 1** in the acceleration distribution of each species. Each squared instantaneous acceleration is mapped in the ternary plot coordinates through the relative contribution of $V^2$, $D^2$ and $\epsilon^2$, and this point cloud is approximated in the ternary plot coordinates with a gaussian kernel to display the point distributions. The 0.05, 0.5 and 0.95 quantile isovalues of these distributions are plotted. (**a**) The entire ternary plot is displayed. The dashed line represents the zoom box represented in Fig. (**b**), where the same isolines are displayed, but with a zoom in in the $y$ direction to highlight differences between species. The number of trajectories are identical than in **Figure 5**: n = 517 (*B. pumilus*), n = 237 (*B. sphaericus*) and n = 279 (*B. cereus*).

## Appendix 3

### Various inference models
Different inference models were designed and tested from the dimensionless state *equation (2)*.

### SSM model
The inference model can be stated as a space-state model (SSM) which is a framework commonly used in spatial ecology to infer a true state, that is true positions and trajectories, and population-wide random walk parameters from time-serie data (*Auger-Méthé et al., 2021*). The SSM inference model is a generalization of Hidden Markov Models (HMM).

Note $z_i^s(t)$ the true (hidden) position of the individual of the species $s$ at time $t$. The state model on acceleration (4) can be rewritten as

$$\frac{d\mathbf{v}_i^s(t)}{dt} = \gamma(v_0^s + b(t, z_i^s(t))(v_1^s - v_0^s) - \|\nu_i^s(t)\|)\frac{\mathbf{v}_i^s(t)}{\|\mathbf{v}_i^s(t)\|} + \beta^s \frac{\nabla b(t, z_i^s(t))}{\|(t, z_i^s(t))\|} + \eta_{\mathbf{mod}}^{\mathbf{s}} \tag{14}$$

$$\frac{z_i^s(t)}{dt} = \mathbf{v}_i^s(t) \tag{15}$$

In this equation, $\mathbf{v}_i^s$ is the true hidden swimmer velocity. Starting from observed initial conditions $z_i^s(0)$, $\mathbf{v}_i^s(0)$, *Equation 16* can be integrated in time to recover hidden $z_i^s(t)$, $\mathbf{v}_i^s(t)$ for all times $t$.

Then, a likelihood equation can be written to compare the true hidden state to the observations.

$$X_i^s(t) \sim z_i^s(t) + \eta_{\mathbf{obs}}^{\mathbf{s}} \tag{16}$$

We note a link between $\eta_{mod}$ and $\eta_{obs}$ in *Equations 15 and 16* and the random state $\eta$ in *Equation 4*. Namely, noting $\sigma_{mod}$ and $\sigma_{obs}$ the standard deviation of the gaussian noises $\eta_{mod}$ and $\eta_{obs}$, direct finite-difference of $A_i^s(t)$ from the true state gives an estimate of the noise variance on the acceleration of the non-linear regression model

$$\epsilon = \sqrt{\left(\frac{\sigma_{mod}}{\Delta t}\right)^2 + \left(\sqrt{6}\frac{\sigma_{obs}}{\Delta t^2}\right)^2}.$$

Compared to problem (5), the main advantages are that the likelihood is written on the original data, that is the observed position, and not a post-processed observed acceleration, subject to finite-difference errors. Furthermore, the true trajectories are recovered and modelling errors $\eta_{\mathbf{mod}}^{\mathbf{s}}$ and observation errors $\eta_{\mathbf{obs}}^{\mathbf{s}}$ are separated. The main drawback of this methodology is that the state space is very large since it includes all the positions and speeds at every time for every swimmers, which leads to intractable computations.

### Mixing SSM and non-linear inference models
An intermediary strategy has been designed by selecting swimmer trajectories that we want to infer by SSM, the remaining trajectories being kept to compute an acceleration dataset $A_i^s(t)$. Namely, note $D_{ssm}$ the set of swimmer index kept for SSM, and $D_A$ the set of swimmer index kept for non-linear regression. We set, for $i \in D_{ssm}$

$$\frac{d\mathbf{v}_i^s(t)}{dt} = \gamma(v_0^s + b(t, z_i^s(t))(v_1^s - v_0^s) - \|\nu_i^s(t)\|)\frac{\mathbf{v}_i^s(t)}{\|\mathbf{v}_i^s(t)\|} + \beta^s \frac{\nabla b(t, z_i^s(t))}{\|(t, z_i^s(t))\|} + \eta_{\mathbf{mod}}^{\mathbf{s}} \tag{17}$$

$$\frac{z_i^s(t)}{dt} = \mathbf{v}_i^s(t) \tag{18}$$

for given initial conditions $z_i^s(0)$, $\mathbf{v}_i^s(0)$, and for $j \in D_A$

$$A_j^s(t) = \gamma(v_0^s + b(t, X_j^s(t))(v_1^s - v_0^s) - \|\nu_i^s(t)\|)\frac{V_j^s(t)}{\|V_j^s(t)\|} + \beta^s \frac{\nabla b(t, X_j^s(t))}{\|\nabla b(t, X_j^s(t))\|} + \eta^{\mathbf{s}} \tag{19}$$

where $X_j^s(t)$, $V_j^s(t)$ and $A_j^s(t)$ are observed positions, speeds and accelerations. This model is completed by a likelihood equation

$$X_i^s(t) \sim z_i^s(t) + \eta_{\mathbf{obs}}^{\mathbf{s}}, \text{ for } i \in D_{SSM} \tag{20}$$

$$A_i^s(t) \quad \sim f_A(\theta^s|b, X_j^s(t), V_j^s(t), A_j^s(t)) + \eta^s \tag{21}$$

where $f_A$ is defined in **equation (4)**.

This setting kept some advantages of the SSM, like inferring some true hidden trajectories or separating the estimate of modeling and observation errors, while limiting the computational load if $D_{SSM}$ is not too large.

We finally kept the regression model for several reasons. First, we are interested in recovering population wide parameters to characterize strain-specific swims, and not identifying true trajectories. Second, we can consider that the observation error with confocal microscopy is several order of magnitudes under the spatial characteristic lengths involved in **equation (2)**, so that observation errors can be neglected. Hence, the objective of separating the uncertainty sources between model and observation errors, which is a main advantage of the SSM or mixed inference settings, becomes secondary. Furthermore, enhancing the state space dimension provided additional uncertainties, worsening the inference precision on synthetic data. We then opted for the simple regression model that provided sufficient parameter identifiability for limited computational load.

## Appendix 4

### KDE computation

We illustrate the process of visualization of multiple point distributions in the same graph using KDE and isolines enclosing specific proportions of the data in *Appendix 4—figure 1*. A point cloud is first approximated with a Gaussian KDE. Then, the value of the gaussian KDE is evaluated in each point of the original point cloud, which allows to map the 2D map into a 1D set where order relation can be defined. Specific quantiles of the resulting values are computed (namely quantile 0.05, 0.5 and 0.95). By definition, the quantile 0.05 separate 5% of the points of the original dataset (the 5% lowest Gaussian KDE values) from the remainder of the data set. The isoline corresponding to the quantile 0.05 then also separates in the 2D map the 5% lowest Gaussian KDE values from the others.

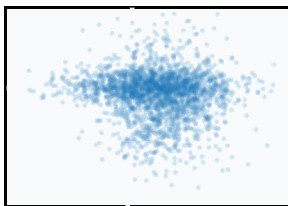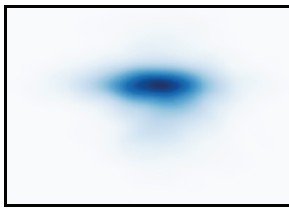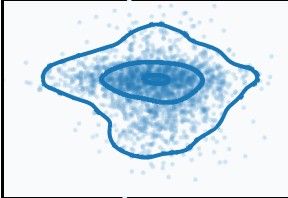

**Appendix 4—figure 1.** Illustration of the Gaussian KDE isovalues computation. Starting from a random 2D point distribution (left panel), a gaussian KDE is computed using the scipy.stats function (middle panel). Then, the gaussian KDE is evaluated at the original point positions, and quantiles of the resulting values are computed (quantiles 0.05, 0.5 and 0.95). Gaussian KDE isolines corresponding to this quantiles are finally computed (right panel). This isolines enclose respectively 5, 50% and 95% of the points of the original distribution, centered in the densest area of the initial point cloud. This procedure gives a good representation of the shape of the data, but allows to display several distributions in the same graph, enabling comparison while avoiding superimposition issues.

