## [Editor Report]

This paper nicely considers how the biofilm matrix impacts the organism's moving within that environment, connecting prior analyses of cell movements on/within abiotic substrates to those within a "living" substrate. Though there are instinctive descriptions for this motility, the strength of this manuscript is the development and implementation of a statistical model that quantifies critical parameters and incorporates interactions with the biofilm matrix itself. While the manuscript measures the differences between morphologically distinct bacteria, a long-term possibility is to achieve predictable and reliable delivery of antimicrobials (delivered by bacteria or an abiotic object) into the biofilm's center, thereby reducing a biofilm's recalcitrant responses to biocontrol chemicals.

---

## [Decision Letter]

**Decision letter after peer review:**

Thank you for submitting your article "Inferring characteristics of bacterial swimming in biofilm matrix from time-lapse confocal laser scanning microscopy" for consideration by *eLife*. Your article has been reviewed by 3 peer reviewers, one of whom is a member of our Board of Reviewing Editors, and the evaluation has been overseen by Wendy Garrett as the Senior Editor. The following individual involved in review of your submission has agreed to reveal their identity: Iago Grobas (Reviewer #2).

Essential revisions:

Overall, the presented data support the conclusions and provide insights into bacterial motility. However, there are concerns about the model (assumptions and extensions), the limited discussion/application of bacterial motility and morphology, and aspects of the experimental design. If you choose to revise this manuscript, please address the following items:

1) In its current form, the paper lacks a characterization of the swimming motility of the bacteria in a Newtonian fluid, an important aspect to rationalize the impact of the biofilm.

2) Clearly communicate how this model would help characterize bacteria motility in exogeneous biofilms (instead of performing microscopy).

3) The discussion of cell morphology and potential impacts on motility within the biofilm is inadequate. Cell morphology should also be considered when characterizing swimming motility in a Newtonian fluid.

See the detailed comments for specific concerns regarding each of these items.*Reviewer #1 (Recommendations for the authors):*

This manuscript by Ravel and colleagues connects prior analysis of bacterial movements on/within abiotic substrates to those within a "living" substrate. The primary outcome is developing a methodology to examine the overall swimming of a bacterial population within another bacterium's biofilm. Implementing these methods would allow for differentiation among bacterial species, identifying microbes that can travel more deeply within an existing biofilm. The authors consider that in the long-term, one could potentially use microbes to reliably deliver antimicrobials (or similar) into the heart of a biofilm, thereby hopefully reducing the biofilm's recalcitrant responses to these biocontrol chemicals.

Strengths

– These scientists develop an inference model and supportive methods to ascertain traits of the population of swimming cells. While similar methods and models exist, the specific examination within a "living" biofilm is intriguing and foundational for developing drug delivery methods or interpreting how cells of any type, e.g., immune cells, might penetrate these bacterial communities. These results also raise questions about the role of cell shape, size, and motility in infiltrating already formed colonies, and conversely, how the composition of the extracellular matrix and cell stacking could protect biofilms from invasion.

– The scientists sequentially test their inference model with and without experimental data. Similar results emerge, suggesting that the inference model could provide an initial step to evaluate different bacterial strains with various biofilm conditions rapidly.

– The authors do a good job of clearly explaining each variable's definition and directly addressing many of the assumptions and constraints of these experiments.

Overall, the presented data support the conclusions, except for the discussion about bacterial shape and motility.

Weaknesses

– An underlying thread is that the difference in shape and flagella size/motility contributes to how each Bacillus strain can navigate the *S. aureus* biofilm. Yet, the discussion of these contributions is not until the end and relies on abstract assertions of potential behavioral differences. This analysis could be more robust if it included analysis of each species' motility in the absence of biofilm so as to establish if and how these species swim differently from one another. As such, lines 416 – 419 are not strongly supported by the data in this manuscript or the current literature.

– There is a potential bias in the data due to the constraints of the experimental set-up: only focal planes near the well's edge are included in the analysis. The biofilm in these zones could have a distinct physical structure (due to the well's wall) than the remainder of the biofilm. While the datasets presented here are self-contained for analysis, any conclusions about the general/overall biofilm are more narrow (or should be taken with a caveat).

– The time-scale of 30 minutes at 30{degree sign}C could permit bacterial growth while cells are moving. The inference measurements appear to be reasonably robust against these potential impacts (perhaps incorporated as part of the "noise" variable). The possible cell growth is worth also considering if attributing differential swimming behaviors to cell morphology or flagella size/location.

I have included all of my scientific comments in the public review. A couple of typographical errors alter a sentence's meaning. The critical ones are as follows:

– line 96, dies  dyes

– line 198, γ  β (?)

Further, please clarify that in Table 1 that the video duration is in minutes.

*Reviewer #2 (Recommendations for the authors):*

This work aims to create a workflow to elucidate how the biofilm matrix affect the trajectories of exogeneous swimming bacteria. In principle, this could potentially be interesting to study biofilm spatial organization and also to characterize transport of synthetic particles such as nanoparticles or colloids carrying biocides into the biofilm. However, the statistical model presented here does not consider the surface interactions between the particles and the porous media which makes the model quite specific to the bacteria-biofilm interaction and, therefore, difficult to extend to other particles and porous materials.

The article is mainly focused on the statistical model which quantifies the key parameters of the bacteria swimming motility affecting their trajectory within the biofilm matrix. The main parameters studied are the acceleration, swimming speed, net displacement in the swimming trajectory and the area covered by the bacteria during the trajectory. The variables in the biofilm matrix that can affect those parameters are the local differences in cell density within the biofilm and the cell density itself. In this regard, the authors found a good agreement between the trajectories inferred by the model and the real trajectories, validating their model for the strains tested. The model allows to decipher how the different bacteria species adapt to the biofilm matrix. Nevertheless, I fail to understand how the model provides an advantage with respect to just observing the trajectories under the microscope. Following the bacteria with microscopy could identify when they slow down or speed up depending on the density of the host biofilm matrix as the model does.

Finally, the authors connect the bacteria morphology of the three species studied to the kinematic descriptors of the bacteria trajectories. The morphological features logically agree with what their model predicts, meaning that, long bacteria are slower in the biofilm matrix and bacteria with lower aspect ratio finds easier to go through the porous biofilm matrix. I find this like a nice way of checking that the model works but at the same time I wonder how useful this model can be since the mentioned morphological features and their impact in the bacteria trajectories could be inferred without the model, just by using microscopy. On the same line, the authors suggest that a brush-like group of thin flagella make changing directions easier but I do not see how this is checked experimentally.

In summary, the parameters affecting the model and the correlation between the kinematic variables of the swimming trajectories and the local conditions in the biofilm are thoroughly checked. However, my concern is that I do not see how this model can widen our knowledge about how bacteria navigate biofilm matrix since tracking bacteria in a biofilm matrix would give similar information. Furthermore, I do not know how limited this model is in terms of the surface chemistry interaction between the biofilm and the bacteria, or the particles in general, that are introduced in the biofilm. This surface chemistry interaction could totally change the trajectory of the swimmers independently on the parameters studied in this article which limits the model to bacteria swimming in a host biofilm and does not allow the extension of the model to synthetic particles or other porous media.

I do not understand very well Figure 1a. There are red bacteria (guest bacteria) in the last frame that do not appear in previous timestamps. Specifically, the trajectory at the bottom right has its origin in an area that is visible for all timepoints and I do not manage to see any bacteria in there in any of the timestamps.

I have also a problem with Figure 1b. I guess that in the caption, 'distance' refers to the shortest distance between the initial and final point of the trajectory. But this is what the authors seem to be drawing in the figure for 'displacement' which is defined as 'the total length of the trajectory path'. So I think these two should be swapped. Actually, I think the correct representation according to the capture is as drawn in Figure 1d.

In table 1, could the units of time interval be specified?

In Figure 2, is the whole set of trajectories for each species displayed for 1 batch or for the 3 batches? I think Figure 2, would be easier to follow if the order of the bacteria in (a) matches the order in (b). I think it would also benefit from a rough y axis in the first graph of the upper panel of (b). What are the units in the x axis? If the magnitudes are dimensionless because they are normalized, I think this should be said in the caption, also because dimensionless magnitudes are referred as V* and A* in the main text. The legend in the bottom panel (b) containing the names of the bacteria should be in italics and 'B' separated from the rest of the name. There is a mistake in the caption (2b), the magnitude 'Area' is not mentioned in the caption.

Do the authors know why the distributions in Figure 2 look quite similar for B. pumilus and B. cereus but different to B. sphaericus? It seems to me that this might be an artefact coming from the fact that less trajectories are plotted for B. sphaericus. I think it would be informative if they put a label with the number of total trajectories displayed in this panel.

I do not see very well the added value of the bottom panel in Figure 2b. I have read the Results section just looking at the 50% line, I don't think the others present much more additional data.

I think the penalization coefficient 'γ' being inversely proportional to the relaxation time should be further explained.

In line 316 says that B. pumilus shows the highest v0 value, indicating a higher ability to swim fast in low density. But v0 was defined as the speed at the highest density, so this means that B. pumilus swims fast at high density. Which one is right?

I do not know if it can be argued that B. sphaericus presents no difference between v0 and v1 as it is written in the main text. It seems that the v0 is extremely low, almost 0, indicating that it cannot swim at high cell density. However, this strain has the highest v1 mean.

Appendix 1: In Figure 1. I think the red channel's brightness should be increased. When the authors say 'rescalled biofilm density map'? what do they mean? Is it just the first image of the biofilm for a certain condition? Or do they alter the image in any way? In this same Figure 1, there is a white space in the gray frame enclosing the 'swimmer trajectories in normalized biofilm'.

I would put figure 6 in an appendix. I do not think it adds much value to the aim of the paper.

Not sure about Fig7. I think it's better to put a square in the zoomed in regions instead of the dashed line. For example, in the middle panel, the zoomed in version of B. pumilus seems like there are two bacterial bodies attached to each other. However, this is very difficult to see in the zoomed out version. I think this would be clearer if instead of a dashed line there was a dashed square around the region.

Why is there a huge change in the epsilon coefficient depending on the species? I thought random noise would affect roughly the same to all species.

I think in general, the paper would benefit if the figure 7 was changed to figure 1. I would motivate the paper by saying that there are three species with different hydrodynamic properties because of shape, number/type of flagella, etc. and this leads to the hypothesis that the behaviour would be different in the porous biofilm matrix. For example, the longer body of B. sphaericus would be an impediment when navigating the biofilm and therefore they expect lower motility, etc. As the paper is structured now, the whole story seems to be around checking whether their model is correct.

I do not understand the calculation of the visited area. What is the parameter k and ns? Why do they have those values? Why was the visited area not calculated using the trajectory multiplied by the size of the particle?

*Reviewer #3 (Recommendations for the authors):*

Ravel et al., investigate the swimming behavior of three different Bacillus species in the biofilm formed by *Staphylococcus aureus*. The biofilm structure and the swimmers' behavior are experimentally characterized using time-lapse confocal microscopy. The swimmers' behavior is described by several parameters measured from tracking their positions in time, including acceleration and speed. The biofilm structure is described in terms of the density of the biomass. The data highlight differences in the shape of the trajectories, speed distributions, and tendency to visit high-density areas of the biofilm in the three different species. These observations are reproduced using a generative model of the data and tentatively explained in terms of bacterial morphology. The model could, in principle, be used to predict the motility of other bacterial species in biofilms.

The conclusions of this paper are supported by the data and the model. Still, the explanation of the former in terms of bacterial motility and morphology needs to be extended and some aspects of the model need to be clarified.

1) The authors explain the different swimming behavior observed in the biofilm in terms of different bacterial morphology, i.e. shape and number of flagella. However, the impact of these differences in a Newtonian fluid should be evaluated and used to better understand the adaptation strategies to the biofilm. In addition, the observations should be commented in light of the literature studying bacterial motility in non-Newtonian fluids (Phys. Rev. Fluids, 5, 073103 (2020); Nat. Phys, 15, 554-558, (2019); Sci. Rep., 5, 15761 (2015); PNAS, 111, 17771-17776 (2014)), where the impact of the flagellar shape and motility are discussed.

2) While a lot of attention is dedicated to describing how bacterial motility is quantified, very few details are given about the measurement of biofilm density and the possible error sources. Since this is a primary ingredient for the interpretation of the results, I would recommend commenting on the procedure and the sensitivity of density measurement.

3) The authors suggest that bacterial motility could help create channels into the biofilm, affecting transport. Is this effect observed in the experiments? This aspect should be clarified and commented on.

4) The purpose of the model and its pros and cons should be discussed more clearly for a general audience. In particular, the authors should clarify the added value of the model with respect to the experiments.

5) Do these bacteria perform a run-and-tumble or a run-and-reverse type of motion in water? Could a back-and-forth trajectory be described as a run-and-reverse?

6) Table 1: The units are missing.

7) Line 125: It would be helpful if some numbers were given in the text to quantify the "large swimming distances" or the "widest swimming distribution".

8) Figure 1: The biofilm density map should be added in panel a. I would recommend avoiding the superimposition of the distributions of ||A||, ||V||,dist, disp and area in panel b.

9) Line 136: Could the term "few" be quantified? From Figure 1a the exceptions seem to be several.

10) Line 192: The bio-readership may not find clear the term "basal" used to identify the simulations. The same applies to "ground truth" in the following. These terms should be defined to improve clarity.

11) Line 303-306: The conclusion on the variability of the outputs should be clarified. Is it a positive feature or is the model missing something?

---

## [Author Response]

Essential revisions:Overall, the presented data support the conclusions and provide insights into bacterial motility. However, there are concerns about the model (assumptions and extensions), the limited discussion/application of bacterial motility and morphology, and aspects of the experimental design. If you choose to revise this manuscript, please address the following items:

We warmly thank the reviewers for this positive comment. Corrections are detailed below.

1) In its current form, the paper lacks a characterization of the swimming motility of the bacteria in a Newtonian fluid, an important aspect to rationalize the impact of the biofilm.

New experiments have been conducted to image the swimming motility of the studied strains in a Newtonian fluid. We took exactly the same culture protocol for the swimmers, but put them in the buffer which is a Newtonian fluid in the absence of host biofilm. Images have been obtained and post-processed with the same dataflow as previous experiments. We then obtained a control experiment characterizing the microbial swim without biofilm.

We first computed the same trajectory descriptors as for the trajectories inside the biofilm, allowing for pairwise comparisons between species swimming in the control buffer, but also to compare the free swim (control) with the swim in a crowded environment (biofilm). We also conducted an additional characterization by addressing a potential run-and tumble or run-and-reverse behaviour of the species (see Reviewer 3 remark): we computed for each trajectory time point the swimmer speed (defined as the mean between the incoming and outgoing velocity vectors (ǁV_i_^s^(t)ǁ+ǁ V_i_^s^(t-Δt)ǁ)/2, for t∊ (Tis+1, Tend,is)) versus the direction change defined as the angle between the incoming and outgoing velocity vectors θis(t)=arccos⁡((Vis(t)⋅Vis(t−Δt))/(||Vis(t)||Vis(t−Δt)||)). An additional figure was added illustrating run-and-tumble in the host biofilm or in the control buffer.

These additional data provided very interesting insights. We could in particular observe that B.sphaericus has a similar run-and-reverse behaviour in the biofilm or the control buffer, the biofilm strongly impairing the swimming speed and increasing the number of reverse events whereas B.pumilus switches from rectilinear runs in the Newtonian buffer to a pronounced run-and-reverse behaviour in the biofilm with decreased speeds.

By contrast, B.cereus is weakly impacted by the host biofilm: it manages to keep approximately straight trajectories and comparable swimming velocities in the biofilm. Interestingly, the number of reverse events is even reduced in the biofilm.

– line 466 : Material and Methods, imaging protocol for the control experiments in the Newtonian fluid is added.

– Figure 2 of the initial submission has been split in two figures now numbered figures 3 and 5. In Figure 3, a panel has been added with the swimmer trajectories in the control buffer. In Figure 5, the descriptor distribution in the control buffer has been added. The figure interpretations has been modified accordingly in lines 110, 122, 133, 140, 165, 167, 169 and 173.

– A new figure (Figure 4) has been added to assess potential run-and tumble behaviours. The figure description has been added in lines 133. Precise definition of angle and average velocity between two consecutive speed vectors is given line 516 in the Material and Methods.

2) Clearly communicate how this model would help characterize bacteria motility in exogeneous biofilms (instead of performing microscopy).

We thank the reviewers for this fundamental remark. We agree with the reviewers, and in particular with Reviewer 2 who was sceptical about the need of an additional mathematical model, that a large part of the present study is made with direct observations of the trajectories without additional modelling: Figures 3, 4 and 5 of the present submission are nothing else than the extraction of direct descriptors from the microscopy images. In particular, these descriptors allow to quantify swimming characteristics (Figure 5, top panel) but also to link the host biofilm (density and gradient) to the swimmer behaviour (speed and acceleration) (Figure 5, lower panel) or to assess run and tumble (Figure 4).

However, the main goal of this study was to go beyond these descriptive analysis. First, we propose an integrative model able to analyse simultaneously heterogeneous data (density maps and trajectories) and to gather all the swimming behaviours that are considered (speed selection, direction changes and random walk) in the same quantification in order to measure their relative impact on the swimmer trajectories. This method is comparable to ANOVA-like multivariate analysis: parametric phenomelogical mappings are defined to link explicative co-variables to a phenomena (i.e. for example the function defining speed selection from biofilm density) and gathered in the same inference problem. Compared to direct observations, the model we propose additionally tells us for example that speed adaptation is preponderant compared to direction selection (Appendix 2 Figure 4). Secondly, we infer and validate a generative model able to reproduce observed trajectories. This model can be used in future works to predict the impact of a bacterial swimmer on the vascularisation of the host biofilm.

– Section title has been changed line 192 to better communicate that the mathematical model is actually a data analysis method. In the same paragraph, comments were added lines 220.

– Additional sentence is added lines 257 and 322 to clarify the fact that the measurement of the relative contribution of the different swimming mechanisms to the acceleration variance is allowed by the mathematical model.

– Additional discussion has been added line 369 to clearly communicate that the model inference can be seen as an analysis method allowing for multi-data integration and co-analysis.

3) The discussion of cell morphology and potential impacts on motility within the biofilm is inadequate. Cell morphology should also be considered when characterizing swimming motility in a Newtonian fluid.

A new discussion has been added linking the cell morphology to the swimming path in the control Newtonian buffer, but also discussing these results respectively to the literature, as suggested by Reviewer 3.

– The result section was rewritten and made more accurate by linking the presence of multiple flagella to a run-and-tumble behaviour, as observed in other flagellated bacteria such as *E. coli*, line 352.

– A new section has been added in the discussion to put the results in perspective, regarding previous studies of the literature and the new control experiments in the Newtonian buffer, line 447.

See the detailed comments for specific concerns regarding each of these items.Reviewer #1 (Recommendations for the authors):This manuscript by Ravel and colleagues connects prior analysis of bacterial movements on/within abiotic substrates to those within a "living" substrate. The primary outcome is developing a methodology to examine the overall swimming of a bacterial population within another bacterium's biofilm. Implementing these methods would allow for differentiation among bacterial species, identifying microbes that can travel more deeply within an existing biofilm. The authors consider that in the long-term, one could potentially use microbes to reliably deliver antimicrobials (or similar) into the heart of a biofilm, thereby hopefully reducing the biofilm's recalcitrant responses to these biocontrol chemicals.Strengths– These scientists develop an inference model and supportive methods to ascertain traits of the population of swimming cells. While similar methods and models exist, the specific examination within a "living" biofilm is intriguing and foundational for developing drug delivery methods or interpreting how cells of any type, e.g., immune cells, might penetrate these bacterial communities. These results also raise questions about the role of cell shape, size, and motility in infiltrating already formed colonies, and conversely, how the composition of the extracellular matrix and cell stacking could protect biofilms from invasion.– The scientists sequentially test their inference model with and without experimental data. Similar results emerge, suggesting that the inference model could provide an initial step to evaluate different bacterial strains with various biofilm conditions rapidly.– The authors do a good job of clearly explaining each variable's definition and directly addressing many of the assumptions and constraints of these experiments.Overall, the presented data support the conclusions, except for the discussion about bacterial shape and motility.

We thank Reviewer 1 for his positive and encouraging comments. Remarks about the discussion about bacterial shape and motility have been addressed, as detailed in the previous section.

– Results and discussion of the links between shape and motility have been rewritten, see lines 352 and 447.

Weaknesses– An underlying thread is that the difference in shape and flagella size/motility contributes to how each Bacillus strain can navigate the *S. aureus* biofilm. Yet, the discussion of these contributions is not until the end and relies on abstract assertions of potential behavioral differences. This analysis could be more robust if it included analysis of each species' motility in the absence of biofilm so as to establish if and how these species swim differently from one another. As such, lines 416 – 419 are not strongly supported by the data in this manuscript or the current literature.

As suggested by Reviewers 1 and 2 (see below), we moved the ultrastructural microscopy to the beginning of the result section, the biological interpretation of the swimming changes being devolved to the last result section, which has been re-written (see Point 3 above). This section title has been changed accordingly. Lines 416-419 of the initial submission have been reformulated.

– The image showing the bacterial physiology has been moved to the beginning of the results and additional descriptions of the bacteria have been included, line 102 to 107.

– Result subsection title has been changed, line 329.

– Lines 416-419 of the initial submission have been reformulated, line 425.

– There is a potential bias in the data due to the constraints of the experimental set-up: only focal planes near the well's edge are included in the analysis. The biofilm in these zones could have a distinct physical structure (due to the well's wall) than the remainder of the biofilm. While the datasets presented here are self-contained for analysis, any conclusions about the general/overall biofilm are more narrow (or should be taken with a caveat).

We performed an additional analysis to check that the physical structure of the biofilm is not drastically changed near the well’s edge, at the bottom of the well, where the focal plan has been chosen to screen the swimmers. To provide a quantitative analysis, we took 4 replicates of *S. aureus* biofilms that were imaged in 3D using a stack of 6 horizontal images, starting from z = 0 near the well’s edge, to z = 6∆z, at the interface between the biofilm and the bulk solution. To study the between and within biofilm variability in the horizontal images, we subsampled them with a regular Cartesian 4x4 grid, resulting in a 4*6*(4*4)=384 2D images database supplemented by metadata (stack, z and x−y coordinate of the subsample). Pairwise correlation of the biofilm density in the 384 samples has been computed (scikit-learn pairwise distances, ’correlation’ parameter), and the resulting similarity matrix has been displayed using Seaborn package (clustermap) after hierarchical clustering (scipy.cluster.hierarchy linkage function) and added to the annex. Additional permanova has been computed to assess the significance of between-group dissimilarities (skbio.stats.distance permanova function). This study shows that the z direction does not structure the data, contrary to the x−y and stack groups, indicating that the biofilm structure is not different near the well’s bottom edge.

– A description of this additional study has been added to the Material and Method, lines 492 and 628.

– A new section has been added in the annex 1, showing the results of the study, in Annex 1 Figure 2, and in the relative description.

– The time-scale of 30 minutes at 30{degree sign}C could permit bacterial growth while cells are moving. The inference measurements appear to be reasonably robust against these potential impacts (perhaps incorporated as part of the "noise" variable). The possible cell growth is worth also considering if attributing differential swimming behaviors to cell morphology or flagella size/location.

– The video duration was actually of 30 s. We really apologize for this misunderstanding induced by the fact that we forgot to mention the time unit in Table 1 which collects all the dataset characteristics, as highlighted by the three Reviewers. The duration was mentioned in the Material and Methods, but we rewrote the unit in plain letter in order to avoid any ambiguity. This time scale is relevant for motion screening and allow to reduce the number of division events in the dataset. Biofilm colonization at longer time scales has been studied for example in [Houry13088], but the influence of swimmer division in the process is interesting: it could be adressed in future works.

– We indicated the time unit in Table 1 and its caption.

– Video duration was indicated in plain letter in the Material and Methods line 478.

I have included all of my scientific comments in the public review. A couple of typographical errors alter a sentence's meaning. The critical ones are as follows:– line 96, dies  dyes– line 198, γ  β (?)Further, please clarify that in Table 1 that the video duration is in minutes.

Corrections have been included.

– Modifications were made lines 111 and Appendix 1.

– units have been clarified in Table 1 and line 478 but are in seconds.

Reviewer #2 (Recommendations for the authors):This work aims to create a workflow to elucidate how the biofilm matrix affect the trajectories of exogeneous swimming bacteria. In principle, this could potentially be interesting to study biofilm spatial organization and also to characterize transport of synthetic particles such as nanoparticles or colloids carrying biocides into the biofilm.

We thank Reviewer 2 for this description that fits the objectives of our work

However, the statistical model presented here does not consider the surface interactions between the particles and the porous media which makes the model quite specific to the bacteria-biofilm interaction and, therefore, difficult to extend to other particles and porous materials.

We agree with Reviewer 2 that biofilm materials are very complex, and can be related to a limited extent to polymeric fluids or porous materials. We added a discussion of these links and limitations and put our results in perspective relatively to other results on flagellated bacteria in polymeric fluids, as also suggested by Reviewer 3 (see below). Cell-to-cell interactions are an important determinant of the kinematic of the swimmer, but taking it into account would necessitate a precise characterization of the rheology of the host biofilm. We opted for a phenomenological description of the kinematic by decomposing the forces exerted on the swimmer into 3 blocks gathering respectively the net forces involved in the velocity selection, in the direction selection, and in random forces. This approach does not pretend to accurately describe the mechanics at the swimmer surface, but rather to gather information into interpretable blocks, the importance of which is given by the inference. This approach is comparable to ANOVA methods by its rationale.

– A discussion has been provided lines 447.

– The discussion on the overall mathematical approach has been enhanced line 369

The article is mainly focused on the statistical model which quantifies the key parameters of the bacteria swimming motility affecting their trajectory within the biofilm matrix. The main parameters studied are the acceleration, swimming speed, net displacement in the swimming trajectory and the area covered by the bacteria during the trajectory. The variables in the biofilm matrix that can affect those parameters are the local differences in cell density within the biofilm and the cell density itself. In this regard, the authors found a good agreement between the trajectories inferred by the model and the real trajectories, validating their model for the strains tested. The model allows to decipher how the different bacteria species adapt to the biofilm matrix. Nevertheless, I fail to understand how the model provides an advantage with respect to just observing the trajectories under the microscope. Following the bacteria with microscopy could identify when they slow down or speed up depending on the density of the host biofilm matrix as the model does.

We thank Reviewer 2 for this description that we endorse. Answers to the concerns about the need for a mathematical model have been provided above in Section 2, point 3. We hope that these modifications will fit his/her expectations.

– See above Section 2, point 3.

Finally, the authors connect the bacteria morphology of the three species studied to the kinematic descriptors of the bacteria trajectories. The morphological features logically agree with what their model predicts, meaning that, long bacteria are slower in the biofilm matrix and bacteria with lower aspect ratio finds easier to go through the porous biofilm matrix. I find this like a nice way of checking that the model works but at the same time I wonder how useful this model can be since the mentioned morphological features and their impact in the bacteria trajectories could be inferred without the model, just by using microscopy. On the same line, the authors suggest that a brush-like group of thin flagella make changing directions easier but I do not see how this is checked experimentally.

As detailed above in Section 2, point 3, the mathematical model is a way to integrate in the same quantification the interaction between the host biofilm and the swimmer velocity, direction and ”randomness”, in order to identify their relative impact on the overall swimmer trajectories. It can be seen as a specific statistical treatment of the microscopy images.

– Additional discussion of the links between kinematics and morphology has been added as indicated in Section 2, point 4 above.

In summary, the parameters affecting the model and the correlation between the kinematic variables of the swimming trajectories and the local conditions in the biofilm are thoroughly checked. However, my concern is that I do not see how this model can widen our knowledge about how bacteria navigate biofilm matrix since tracking bacteria in a biofilm matrix would give similar information. Furthermore, I do not know how limited this model is in terms of the surface chemistry interaction between the biofilm and the bacteria, or the particles in general, that are introduced in the biofilm. This surface chemistry interaction could totally change the trajectory of the swimmers independently on the parameters studied in this article which limits the model to bacteria swimming in a host biofilm and does not allow the extension of the model to synthetic particles or other porous media.

A comment about the generalization of our work has been added in the discussion

– A comment has been added line 449.

I do not understand very well Figure 1a. There are red bacteria (guest bacteria) in the last frame that do not appear in previous timestamps. Specifically, the trajectory at the bottom right has its origin in an area that is visible for all timepoints and I do not manage to see any bacteria in there in any of the timestamps.

Thank you for this remark. The figure has been remade.

– Figure 1a has been remade with a correct trajectory.

I have also a problem with Figure 1b. I guess that in the caption, 'distance' refers to the shortest distance between the initial and final point of the trajectory. But this is what the authors seem to be drawing in the figure for 'displacement' which is defined as 'the total length of the trajectory path'. So I think these two should be swapped. Actually, I think the correct representation according to the capture is as drawn in Figure 1d.

Thanks for pointing to this concern. The correct definition of distance and displacement was given in their mathematical definition line 135 of the revised version of the manuscript. We corrected accordingly Figure 1.d and its caption.

– Images have been swapped in image 1.d consistently with figure 1.b and mathematical definition.

– Caption has been modified.

In table 1, could the units of time interval be specified?

This is done.

– The unit is added in the corresponding column header

In Figure 2, is the whole set of trajectories for each species displayed for 1 batch or for the 3 batches? I think Figure 2, would be easier to follow if the order of the bacteria in (a) matches the order in (b). I think it would also benefit from a rough y axis in the first graph of the upper panel of (b). What are the units in the x axis? If the magnitudes are dimensionless because they are normalized, I think this should be said in the caption, also because dimensionless magnitudes are referred as V* and A* in the main text. The legend in the bottom panel (b) containing the names of the bacteria should be in italics and 'B' separated from the rest of the name. There is a mistake in the caption (2b), the magnitude 'Area' is not mentioned in the caption.Do the authors know why the distributions in Figure 2 look quite similar for B. pumilus and B. cereus but different to B. sphaericus? It seems to me that this might be an artefact coming from the fact that less trajectories are plotted for B. sphaericus. I think it would be informative if they put a label with the number of total trajectories displayed in this panel.I do not see very well the added value of the bottom panel in Figure 2b. I have read the Results section just looking at the 50% line, I don't think the others present much more additional data.

In previously figure 2 and now figure 3, the 3 batches are pooled. The y axis is not indicated since the distribution are normalized to sum to one. The x-axis are also dimensionless since they are normalized: we added a note in the figure caption. We modified the figures accordingly to the comments.

In our opinion, similar distributions for B.pumilus of B.cereus could come from similar trajectory characteristics for these macroscopic descriptors. We can not discard that discrepancies could come from different sample numbers. The artefact could be discarded by the fact that the number of trajectories for B.sphaericus is in the same order than B.cereus, although they show different descriptor distributions.

The bottom panel in the previously Figure 2b but now figure 3 shows the dependence of the acceleration and speed to the underlying host biofilm structure (density and density gradient). It is a way to present the result of direct observation by microscopy of the links between kinematic characteristics and the host biofilm.

– We agree that most of the information is given by the 50% isoline. We kept however the different lines to comply with eLife policy in distribution presentation, since the authors are asked to present the distribution spread in addition to means or medians.

– Plot order has been modified.

– Names of bacteria in the figure legend have been changed.

– Figure 3 caption has been amended with a note indicating that the 3 batches are pooled.

– The number of samples in the different densities has been indicated in the legend.

I think the penalization coefficient 'γ' being inversely proportional to the relaxation time should be further explained.

The fact that γ is dimensioned as an inverse of time is necessary to be homogeneous to the acceleration in the left hand side of the equation. The term ”relaxation” comes from ”penalization” modelling in fluid mechanics, where given vector fields can be prescribed by adding to the state equation a penalization term penalizing the difference to the prescribed vector fields. The term ”relaxation” refers to the time needed for the computed vector field to relax towards the constraint.

– Additional explanations have been added line 204.

In line 316 says that B. pumilus shows the highest v0 value, indicating a higher ability to swim fast in low density. But v0 was defined as the speed at the highest density, so this means that B. pumilus swims fast at high density. Which one is right?

We corrected the model description to clearly state that v0 is the prescribed speed when b = 0 and v1 when b = 1. The corresponding sentence line 316 was correct.

– The mathematical model is more clearly defined with additional explanation line 204.

I do not know if it can be argued that B. shaericus presents no difference between v0 and v1 as it is written in the main text. It seems that the v0 is extremely low, almost 0, indicating that it cannot swim at high cell density. However, this strain has the highest v1 mean.

The sentence has been reformulated to be more accurate. The same sentence was repeated later in the original submission: it has been also modified.

– Modifications have been added lines 303 and 317.

Appendix 1: In Figure 1. I think the red channel's brightness should be increased. When the authors say 'rescalled biofilm density map'? what do they mean? Is it just the first image of the biofilm for a certain condition? Or do they alter the image in any way? In this same Figure 1, there is a white space in the gray frame enclosing the 'swimmer trajectories in normalized biofilm'.

The figure has been remade. Indications have been added in Annex 1 to state that brightness has been changed, and to recall the Material and method sentence that detail the rescaling applied to the biofilm density map. No other post-processing has been applied to the raw biofilm maps than mapping in gray scale and color rescaling between 0 and 1.

– The figure has been remade with increased brightness.

I would put figure 6 in an appendix. I do not think it adds much value to the aim of the paper.

This figure shows the balance between the different swimming processes. It shows consistency with the other results. In particular, B.cereus is the bacteria which is the most influenced by the biofilm gradients. It is also the bacteria with the highest stochastic term, related to straight lines in the Langevin equation. These data are consistent with a higher hability of B.cereus to navigate in the biofilm and conserve straight runs.

– Figure 6 has been moved to Annex 2. Additional text has been added for the annex to be self-contained.

Not sure about Fig7. I think it's better to put a square in the zoomed in regions instead of the dashed line. For example, in the middle panel, the zoomed in version of B. pumilus seems like there are two bacterial bodies attached to each other. However, this is very difficult to see in the zoomed out version. I think this would be clearer if instead of a dashed line there was a dashed square around the region.

The zoomed in images were not obtained by digital zoom in but optical zoom in. The bacteria of zoomed in image do not correspond to the bacteria of the larger scale image. That is why we did not opt for a dashed square around the region that was zoomed in.

– We added a complementary sentence in the zoomed in legend to explain that fact.

Why is there a huge change in the epsilon coefficient depending on the species? I thought random noise would affect roughly the same to all species.

The huge change in the epsilon mainly come from the fact that the mechanistic part of the model do not explain the observed kinematics to the same extent, which is reflected in the different goodness of fit for the 3 bacteria (see Table 2). Note however that a larger stochastic part is also related to straighter trajectories (see Figure 1 and Annex 2 Figure 1): the higher stochastic part of B.cereus can be related to its straighter runs in the data (see Figure 3).

I think in general, the paper would benefit if the figure 7 was changed to figure 1. I would motivate the paper by saying that there are three species with different hydrodynamic properties because of shape, number/type of flagella, etc. and this leads to the hypothesis that the behaviour would be different in the porous biofilm matrix. For example, the longer body of B. sphaericus would be an impediment when navigating the biofilm and therefore they expect lower motility, etc. As the paper is structured now, the whole story seems to be around checking whether their model is correct.

We thank Reviewer 2 for this advice. We moved the ultimately figure 7 to the top of the paper. We added a brief description of the ultrastuctural bacterial morphology and linked that section to the remainder of the article by checking if these physiological discrepancies could give the bacteria different swimming behaviour in the biofilm matrix. See also above (Reviewer 1’s public review, point 2).

– We moved the image and added a new paragraph lines 102 and 107.

I do not understand the calculation of the visited area. What is the parameter k and ns? Why do they have those values? Why was the visited area not calculated using the trajectory multiplied by the size of the particle?

The visited area was not calculated as the trajectory multiplied by the size of the particle to take into account bacteria that visit several times the same zones. This metric would be a proxy of the biofilm vascularization induced by the swimmer : a swimmer that cross several times the same area does not dig additional pores in the host biofilm and does not increase the biofilm vascularization. We then opted for tagging all the pixels of the biofilm image that are crossed by a swimmer. But the displacement of the swimmers during a time step can be higher than several pixels : we then had to reconstruct the trajectory during consecutive swimmer positions to recover the pixels crossed by the swimmer during the time step. As a first order approximation of the trajectory during a time step, we computed a straight line between two successive positions.

The parameter n_s_ was chosen so that ‖Xis(t+Δt)−Xis(t)‖/ns was always inferior to a pixel size. k is not a parameter but a variable varying from 1 to n_s_.

Reviewer #3 (Recommendations for the authors):Ravel et al., investigate the swimming behavior of three different Bacillus species in the biofilm formed by *Staphylococcus aureus*. The biofilm structure and the swimmers' behavior are experimentally characterized using time-lapse confocal microscopy. The swimmers' behavior is described by several parameters measured from tracking their positions in time, including acceleration and speed. The biofilm structure is described in terms of the density of the biomass. The data highlight differences in the shape of the trajectories, speed distributions, and tendency to visit high-density areas of the biofilm in the three different species. These observations are reproduced using a generative model of the data and tentatively explained in terms of bacterial morphology. The model could, in principle, be used to predict the motility of other bacterial species in biofilms.The conclusions of this paper are supported by the data and the model. Still, the explanation of the former in terms of bacterial motility and morphology needs to be extended and some aspects of the model need to be clarified.

We thank Reviewer 3 for this positive comments.

1) The authors explain the different swimming behavior observed in the biofilm in terms of different bacterial morphology, i.e. shape and number of flagella. However, the impact of these differences in a Newtonian fluid should be evaluated and used to better understand the adaptation strategies to the biofilm. In addition, the observations should be commented in light of the literature studying bacterial motility in non-Newtonian fluids (Phys. Rev. Fluids, 5, 073103 (2020); Nat. Phys, 15, 554-558, (2019); Sci. Rep., 5, 15761 (2015); PNAS, 111, 17771-17776 (2014)), where the impact of the flagellar shape and motility are discussed.

We thank Reviewer 3 for pointing out these references. We added additional discussions, in light of these studies.

– Additional discussion added line 447.

2) While a lot of attention is dedicated to describing how bacterial motility is quantified, very few details are given about the measurement of biofilm density and the possible error sources. Since this is a primary ingredient for the interpretation of the results, I would recommend commenting on the procedure and the sensitivity of density measurement.

We investigated the effect on parameter estimate of increasing noise on the biofilm data. The impact of noise on the parameter inference is assessed by noising the biofilm density and the biofilm density gradients with an additive gaussian noise with increasing variance. The noise variance is scaled with the variance observed in the original data. Namely, we set(1)∈b∼N(0,l∗σb)

and(2)ϵ∇b∼N(0,2lΔx∗σb)

where σ_b_ is the variance observed in the original data, and b and ∊_b_ and ∊_∇b_are respectively the noise applied to the biofilm density and the biofilm density gradient. The parameter l ∈ [0,0.01,0.02,0.03,0.04,0.05] is increased to apply a noise from 0 to 5%.

The parameter β is the most impacted (error of 35% for 5% noise). But it is also the parameter that has the minimal influence on the swimmer model when no noise is applied, showing a lower identifiability for this parameter. For the other parameters, the impact is limited.

– A new figure and its description is added in Annex 1 to show the impact of increasing noise on the parameter estimate.

– Sensitivity analysis method is detailed in Material and method line 591.

– A reference to the figure is added Appendix 2.

3) The authors suggest that bacterial motility could help create channels into the biofilm, affecting transport. Is this effect observed in the experiments? This aspect should be clarified and commented on.

This effect was observed during this experiment but not quantified. The impact of swimmer as facilitator of the penetration of macromolecules inside the biofilm has been published by co-authors in [Houry13088].

– A comment has been added line 54.

– A new image has been added (Annex 1, figure 3) to illustrate the pore formation.

4) The purpose of the model and its pros and cons should be discussed more clearly for a general audience. In particular, the authors should clarify the added value of the model with respect to the experiments.

This comment has been addressed (see essential Revisions above).

– see Section 2 point 3 above.

5) Do these bacteria perform a run-and-tumble or a run-and-reverse type of motion in water? Could a back-and-forth trajectory be described as a run-and-reverse?

As stated above, we analysed the new experiments in the control Newtonian buffer, together with the data in the biofilm, by searching for run-and-tumble or run-and-reverse episode. We can now state that B.sphaericus performs run-and-reverse swimming in a Newtonian fluid. This behaviour is not changed in a biofilm. Conversely, B.pumilus has a different swimming pattern in a Newtonian fluid (straight lines) and in the biofilm (increase of the number of tumbling). B.cereus has the same type of swimming in the Newtonian buffer or in the biofilm.

– An additional figure has been added (Figure 4), with comments and description (see above).

6) Table 1: The units are missing.

This is corrected

– corrections have been made in the table column header and in the material and methods (see above).

7) Line 125: It would be helpful if some numbers were given in the text to quantify the "large swimming distances" or the "widest swimming distribution".

Additional quantifications were provided.

– Value for average speed and acceleration were given: line 154 and 155

– Large swimming distances have been documented with numbers: line 142 and 142

– The widest distribution has been documented with numbers: line 145

8) Figure 1: The biofilm density map should be added in panel a. I would recommend avoiding the superimposition of the distributions of ||A||, ||V||,dist, disp and area in panel b.

The biofilm density map can hardly be added to panel a since (1) the trajectories of the 3 batches are pooled, corresponding to 3 different density map temporal stacks, (2) the density map is changing overtime due to the mechanical action of the swimmers. In the annex, we presented trajectories with an underlying biofilm density map for illustrative purpose, but we highlighted that the biofilm information is a time-series, and that the swimmer position is associated to the biofilm density at the corresponding time. The distributions were modified.

– Figure 5 has been modified

9) Line 136: Could the term "few" be quantified? From Figure 1a the exceptions seem to be several.

We gave quantification of the fact that few trajectories of B.sphaericus are leaving the neighbourhood of their starting point: we stated that 6% of the B.sphaericus induce a displacement of more than 10 µm compared to 28% for B.cereus and 26% for B.pumilus.

– Additional details are given line 171.

10) Line 192: The bio-readership may not find clear the term "basal" used to identify the simulations. The same applies to "ground truth" in the following. These terms should be defined to improve clarity.

Defintions were added.

– we added definitions for these terms Appendix 2.

– we added additional precisions in the legend of Figure 6.

11) Line 303-306: The conclusion on the variability of the outputs should be clarified. Is it a positive feature or is the model missing something?

The model is recovering the main characteristics of the swimmer population, but is not able to reproduce the whole variability of the data, meaning that it is less accurate at reproducing points in the distribution tail.

– Clarifications are given line 288.